# DuFal: Dual-Frequency-Aware Learning for High-Fidelity Extremely Sparse-view CBCT Reconstruction

**Cuong Tran Van\***                                                         *cuong.anduy@gmail.com*
*FPT Software AI Center, Viet Nam*

**Trong-Thang Pham\***                                                         *tp030@uark.edu*
*AICV Lab, EECS Department, University of Arkansas*

**Ngoc Son Nguyen**                                                         *sonnn45@fpt.com*
*FPT Software AI Center, Viet Nam*

**Duy Minh Ho Nguyen**                                         *ho_minh_duy.nguyen@dfki.de*
*Max Planck Research School for Intelligent Systems (IMPRS-IS), University of Stuttgart, and DFKI, Germany*

**Ngan Le**                                                         *thile@uark.edu*
*AICV Lab, EECS Department, University of Arkansas*

**\*Co-first authors**

**Reviewed on OpenReview:** `https://openreview.net/forum?id=2wAZjAtK16&referrer`

## Abstract

Sparse-view Cone-Beam Computed Tomography reconstruction from limited X-ray projections remains a challenging problem in medical imaging due to the inherent undersampling of fine-grained anatomical details, which correspond to high-frequency components. Conventional CNN-based methods often struggle to recover these fine structures, as they are typically biased toward learning low-frequency information. To address this challenge, this paper presents **DuFal** (Dual-Frequency-Aware Learning), a novel framework that integrates frequency-domain and spatial-domain processing via a dual-path architecture. The core innovation lies in our **High-Local Factorized Fourier Neural Operator**, which comprises two complementary branches: a Global High-Frequency Enhanced Fourier Neural Operator that captures global frequency patterns and a Local High-Frequency Enhanced Fourier Neural Operator that processes spatially partitioned patches to preserve spatial locality that might be lost in global frequency analysis. To improve efficiency, we design a **Spectral-Channel Factorization** scheme that reduces the Fourier Neural Operator parameter count. We also design a **Cross-Attention Frequency Fusion** module to integrate spatial and frequency features effectively. The fused features are then decoded through a Feature Decoder to produce projection representations, which are subsequently processed through an Intensity Field Decoding pipeline to reconstruct a final Computed Tomography volume. Experimental results on the LUNA16 and ToothFairy datasets demonstrate that DuFal significantly outperforms existing state-of-the-art methods in preserving high-frequency anatomical features, particularly under extremely sparse-view settings.

## 1 Introduction

Computed Tomography (CT) plays a vital role in medical imaging, providing clinicians with detailed, non-invasive visualization of internal anatomical structures to support accurate diagnosis and treatment planning. Among CT modalities, Cone-Beam Computed Tomography (CBCT) (Alamri et al., 2012) has gained prominence due to its ability to produce high-resolution 3D images with faster scanning times compared to

traditional Fan-Beam CT (Decabooter et al., 2024). However, achieving high image quality often necessitates higher radiation doses, which raise safety concerns for patients. To mitigate this issue, substantial research has focused on developing low-dose CT imaging protocols, particularly through sparse-view CBCT reconstruction, which reduces radiation exposure by acquiring fewer projection views during scanning.

Traditional reconstruction methods such as Filtered Back Projection (FBP) (Turbell, 2001), Feldkamp Davis Kress (FDK) (Turbell, 2001), and iterative techniques like Algebraic Reconstruction Technique (ART) and Simultaneous Algebraic Reconstruction Technique (SART) (Andersen & Kak, 1984) perform well when dense-view data ($\geq 100$ projections) is available. However, their performance deteriorates significantly as the number of views decreases. In sparse-view (20-50 projections) and especially extremely sparse-view ($\leq 10$ projections) settings, these methods struggle due to the ill-posed nature of the inverse problem, limiting their applicability in low-dose imaging scenarios.

In pursuing extremely sparse-view ($\leq 10$) CBCT reconstruction, which aims to achieve the lowest possible radiation dose, deep learning has emerged as a powerful approach to address the challenges posed by limited projection data. Current sparse-view CBCT reconstruction methods can be categorized into three main approaches. First, neural attenuation field models such as Neural Attenuation Fields (NAF) (Zha et al., 2022b) and SAX-NeRF (Cai et al., 2024) directly model the underlying anatomical structures from X-ray projections. Second, post-processing enhancement methods apply CNN-based architectures to improve low-quality CT reconstructions, including DuDoNet (Lin et al., 2019) for artifact reduction, FBPConvNet (Jin et al., 2017) for filtered back-projection enhancement, and various denoising techniques using encoder-decoder frameworks with skip connections. However, CNN-based models are biased toward low-frequency information, as demonstrated in previous studies (Xu et al., 2020; Magid et al., 2021; Lin et al., 2022; Khayatkhoei & Elgammal, 2022), which hinders their ability to capture high-frequency features essential for preserving fine anatomical details in medical imaging. Third, end-to-end implicit neural representation methods such as DIF-Net (Lin et al., 2023), C2RV-Net (Lin et al., 2024c), and DIF-Gaussian (Lin et al., 2024a) attempt to reconstruct CT volumes directly from X-ray projections without intermediate low-quality CT generation. Despite this direct approach, these methods remain constrained by spatial domain processing limitations that similarly struggle to preserve the high-frequency details essential for medical imaging precision.

Recognizing the importance of capturing high-frequency information, FreeSeed (Ma et al., 2023) attempts a hard fix with a band-pass Fourier module to learn frequency representations, but this approach lacks flexibility because it is hard-coded into each convolutional layer, making this mechanism difficult to seamlessly integrate into other models. Moreover, FreeSeed only operates as a post-processing denoising method on already-reconstructed CT images rather than performing direct reconstruction from raw X-ray data. While global high-frequency processing may effectively capture edges and structural boundaries, small objects such as nodules in medical images require localized patch-wise high-frequency analysis to preserve fine details that are important for detection tasks.

This work introduces the **DuFal** (Dual-Frequency-Aware Learning) framework for sparse-view CBCT reconstruction. DuFal follows a two-stage design: (i) the **Frequency-enhanced Dual-encoding** stage includes a dual-encoder that processes each projection with a standard spatial encoder in parallel with a proposed *Frequency Encoder*, then passes through a Feature Decoder to obtain the necessary features for the next stage. The Frequency Encoder captures complementary global and local spectral cues using multiple *High-Local Factorized Fourier Neural Operator* (HiLocFFNO) layers, while the spatial encoder models structural context in the image domain. And (ii) the **Intensity Field Decoding** stage, which is a standard decoding pipeline from Deep Intensity Field (DIF) that takes the decoded features and reconstructs them into a 3D CT volume.

Since Fourier Neural Operators (FNO) exhibit rapid parameter growth as the number of retained frequency modes increases, we introduce *Spectral-Channel Factorization* (SCF), a weight factorization that substantially reduces parameters while maintaining comparable accuracy. To preserve important spectral information when merging the two pathways, we use *Cross-Attention Frequency Fusion* (CAFF), which operates in the frequency domain to combine spatial queries with frequency keys/values at each resolution level. The fused multi-scale features are decoded and subsequently passed through the DIF pipeline for volumetric reconstruction. Both the Frequency Encoder and CAFF are designed as modular units that can be seamlessly integrated into

existing frameworks. As shown in Table 1, DuFal is, to our knowledge, the first end-to-end method for sparse-view CBCT reconstruction that jointly models spatial features and global-local frequency information.

In summary, our key contributions are as follows:

- We propose **DuFal**, a complete end-to-end framework that integrates a dual-encoding architecture with **Frequency-enhanced Dual-encoding** for sparse-view CBCT reconstruction. Our framework processes X-ray projections through parallel spatial and frequency pathways, enabling simultaneous capture of structural context and high-frequency anatomical details that are often lost in conventional CNN-based approaches.

- To complement the global receptive field of FNO, we introduce **HiLocFFNO** blocks that combine global high-frequency enhancement with local patch-based processing through dual FNO branches. This design preserves fine-grained details at both global and local scales.

- We propose a novel weight decomposition method called **SCF** that factorizes complex FNO weight tensors into separate channel-mixing and spectral-weighting components, achieving 82.20% total parameter reduction compared to using vanilla FNO while maintaining reconstruction quality for high-resolution medical imaging.

- We design a **CAFF** mechanism that operates directly in the frequency domain to integrate spatial and spectral features through cross-attention on real and imaginary components separately. This fusion strategy preserves complementary information from both encoding pathways, especially the frequency characteristics essential for high-quality CT reconstruction.

## 2 Related Work

### 2.1 Sparse-view CBCT Reconstruction

Traditional CT reconstruction methods such as Filtered Back Projection (FDK) (Feldkamp et al., 1984) and Algebraic Reconstruction Techniques (ART) (Andersen & Kak, 1984) require hundreds of projections acquired over a full rotation. When limited to sparse views ($< 100$ views), these methods often produce severe artifacts and degraded image quality. To mitigate this issue, deep learning methods have become popular in sparse-view reconstruction. Post-hoc denoising methods (Han et al., 2016; Jin et al., 2017; Ma et al., 2023; Wu et al., 2022) have been introduced for the reconstruction of conventional fan/parallel-beam CT. For example, FreeSeed (Ma et al., 2023) applies filtered back projection (FBP) to reconstruct artifact-laden CT volumes and designs a specialized architecture to capture frequency information for denoising and refining missing details. However, FreeSeed only captures frequency information at the global level to expand the overall receptive field. Moreover, when adapted to CBCT through slice-wise processing, these methods often fail to maintain the spatial consistency of reconstructed 3D volumes (Lin et al., 2024a).

Advancing to multi-view CBCT reconstruction, several representative approaches utilize only one or two perpendicular X-ray projections (Ying et al., 2019; Corona-Figueroa et al., 2022; Kyung et al., 2023; Liu et al., 2024b; Jeong et al., 2025; Liu & Bai, 2024). Although these techniques demonstrate potential, they are specifically engineered for single/orthogonal-view reconstruction and are difficult to extend to general sparse-view reconstruction scenarios.

For more realistic multi-view configurations, i.e., CBCT reconstruction, a promising approach is to exploit implicit neural representations (INRs), which represent the CT volume as a continuous attenuation field. Self-supervised INR methods such as NAF (Zha et al., 2022b) and NeRP (Shen et al., 2022b) minimize the discrepancy between real and rendered X-ray images without requiring full 3D supervision. However, these self-supervised methods require good priors before per-sample optimization, which makes them unsuitable for extremely sparse views with very limited prior data. Supervised INR methods like DIF-Net (Lin et al., 2023) encode multiple X-ray views with a 2D encoder and use these to map 3D coordinates to latent features in projection space. DIF-Gaussian (Lin et al., 2024a) further improves on DIF-Net by introducing 3D Gaussians as spatial feature carriers, allowing more expressive and geometry-aware reconstructions. However, these

methods heavily depend on 2D encoders, which ignore the fundamental problem of CNNs, their preference for low-frequency information.

Our framework DuFal is a supervised INR method (like DIF-Net and DIF-Gaussian), where we view a CT volume as a continuous attenuation field. However, we address the fundamental limitations of naive spatial encoders by introducing a dedicated Frequency Encoder that can be flexibly integrated with standard spatial encoders and an intuitive cross-attention-based fusion layer. Moreover, our Frequency Encoder captures both global and local frequency information through two different branches that specialize in global and local frequency information, addressing the lack of local frequency information in FreeSeed. Together, DuFal becomes a dual-domain architecture that enables comprehensive feature encoding by simultaneously processing both spatial structural context and frequency domain characteristics of sparse X-ray projections.

Table 1: Capability comparison of existing methods on extreme sparse-view CBCT reconstruction. DuFal captures several key aspects: spatial modeling, end-to-end reconstruction capability, and comprehensive frequency modeling.

| Methods | Spatial Modelling | End-to-end Reconstruction | Frequency Modelling | |
| | | | Global | Local |
| --- | --- | --- | --- | --- |
| FDK (Feldkamp et al., 1984) | ✓ | ✓ | ✗ | ✗ |
| SART (Andersen & Kak, 1984) | ✓ | ✓ | ✗ | ✗ |
| NAF (Zha et al., 2022a) | ✓ | ✓ | ✗ | ✗ |
| NERP (Shen et al., 2022b) | ✓ | ✓ | ✗ | ✗ |
| FBPConvNet (Jin et al., 2017) | ✓ | ✗ | ✗ | ✗ |
| Freeseed (Ma et al., 2023) | ✓ | ✗ | ✗ | ✓ |
| DIF-net (Lin et al., 2023) | ✓ | ✓ | ✗ | ✗ |
| DIF-Gaussian (Lin et al., 2024a) | ✓ | ✓ | ✗ | ✗ |
| **DuFal (Ours)** | ✓ | ✓ | ✓ | ✓ |

## 2.2 Fourier Transform in Medical Image Analysis

The Fourier transform plays a crucial role in medical image analysis due to its ability to represent frequency components, making it valuable for tasks requiring fine detail and structure preservation. In segmentation, spectral methods have been used to enhance boundary sensitivity and perform denoising, as seen in works like Spectformer (Patro et al., 2025) and FourierFormer (Nguyen et al., 2022). In diffusion models, spectral-domain formulations accelerate sampling and improve robustness to noise (Li et al., 2023b;a; Si et al., 2024). For 3D medical reconstruction, Fourier-based positional encodings are widely adopted in neural implicit models to capture high-frequency variations and spatial smoothness (Shen et al., 2022b; Zha et al., 2022b; Tancik et al., 2020; Pedraza Ortega et al., 2007; Liu et al., 2024a).

Despite these advances, Fourier-based techniques in sparse-view CBCT reconstruction have not been well explored. Our work introduces a principled use of FNO tailored for this domain, with a focus on preserving high-frequency details through the HiLocFFNO layer. Unlike prior approaches designed for 3D reconstruction tasks that incorporate frequency-aware attention only at later stages of the network or as post-processing (Ma et al., 2023; Pedraza Ortega et al., 2007; Liu et al., 2024a), our model integrates frequency-domain reasoning into the encoder.

Additionally, we choose the supervised INR framework to utilize its high performance and speed. Note that naively applying FNO significantly reduces our inference speed. We propose the SCF mechanism to improve efficiency through low-rank Fourier weight factorization, which significantly reduces both parameter count and computational overhead. This compact design makes our model well-suited for sparse-view CBCT scenarios where only a limited number of projections are available and computational resources are constrained.

# 3 Preliminary

## 3.1 Problem Definition

Given a set of $K$ sparse 2D X-ray projections $\mathbf{I} = \{I_1, I_2, \ldots, I_K\}$, where each $I_k \in \mathbb{R}^{H_I \times W_I}$ is a grayscale image of height $H_I$ and width $W_I$, and the corresponding gantry angles $\boldsymbol{\alpha} = \{\alpha_1, \alpha_2, \ldots, \alpha_K\}$, the objective

is to learn a function:

$$(\{I_k\}_{k=1}^K, \{\alpha_k\}_{k=1}^K) \mapsto \hat{Y} \in \mathbb{R}^{H_c \times W_c \times D_c} \tag{1}$$

that maps the sparse projection data and their angles to a volumetric CT image $\hat{Y}$. Here, $H_c$, $W_c$, and $D_c$ represent the height, width, and depth of the reconstructed volume, respectively.

### 3.2 Supervised Implicit Neural Representation Framework

Among the existing supervised INR frameworks, we borrow from the DIF framework (Lin et al., 2023), which serves as a straightforward baseline that converts sparse X-ray projections into a continuous intensity field by relying solely on spatial features. In DIF framework, the process involves two major modules: (a) Feature Encoding and (b) Intensity Field Decoding.

**Feature Encoding**. A UNet encoder $\Phi$ processes each projection image $I_k$ and yields the feature tensor $E_k = \Phi(I_k)$, constructing a feature set $\mathcal{E} = \{E_1, E_2, \ldots, E_K\}$ where $E_k \in \mathbb{R}^{C \times H \times W}$. Here, $C$ denotes the number of feature channels, while $H$ and $W$ are the height and width of the spatial resolution in the encoded feature space.

*To fundamentally improve the projection representations, we introduce a novel dual-encoding network,* **Frequency-Enhanced Dual-encoding***, which injects frequency cues into $\mathcal{E}$ prior to decoding. More details are presented in Section 4.*

**Intensity Field Decoding.** This module is implemented in two steps as follows:

- *Step 1: Multi-view Fusion.* This step aims to encode multi-view features $\mathcal{E}$ into a single feature in 3D space. For any given 3D spatial point $x \in \mathbb{R}^3$, we transform it to the 2D coordinate system of each projection view $k$ using the geometric projection function $\varphi(x, \alpha_k, \beta)$, where $\alpha_k$ represents the angle of view $k$ and $\beta$ encompasses the imaging geometry parameters (such as source-to-detector distance, pixel spacing, and detector positioning). This yields the 2D projection coordinates $x'_k \in \mathbb{R}^2$ on the detector panel for view $k$. The corresponding feature representation vector is then extracted through bilinear interpolation: $r_k = \pi(E_k, x'_k) \in \mathbb{R}^C$, where $E_k$ is the feature map and $x'_k$ provides the sampling coordinates. These $K$ view-specific representations $\{r_1, \ldots, r_K\}$ are fused using a set function $\delta$ to obtain the aggregated embedding $\bar{r} = \delta(\{r_1, \ldots, r_K\}) \in \mathbb{R}^C$.

- *Step 2: Intensity Reconstruction.* This step aims to reconstruct the intensity of each 3D point to produce a CT volume. A four-layer MLP $\psi : \mathbb{R}^C \to \mathbb{R}$ maps the fused vector to a continuous intensity prediction $\hat{y}(x) = \psi(\bar{r})$. To generate a full CT volume, we sample a uniform 3D grid with coordinates $x_{ijk} = (i, j, k)$ and evaluate $\hat{y}(x_{ijk})$ over this grid to yield a volumetric output $\hat{Y}$.

*By integrating our proposed Frequency-Enhanced Dual-encoding into the DIF framework, we call it the DuFal framework.* In the proposed DuFal framework, we focus on improving feature encoding within the Feature Encoding module by incorporating fine-grained information while keeping the Intensity Field Decoding module unchanged from the original design.

**Objectives.** Assume the original CT volume has resolution $H_c \times W_c \times D_c$ with physical spacing $(s_h, s_w, s_d)$ in millimeters. During training, instead of supervising the full voxel grid, a set of $N$ 3D points is randomly sampled $X = \{x_1, x_2, \ldots, x_N\}$ in world coordinates, where each $x_n \in \mathbb{R}^3$ is sampled uniformly within the bounding box from $(0, 0, 0)$ to $(s_h H_c, s_w W_c, s_d D_c)$.

The network estimates intensity values $\hat{Y} = \{\hat{y}_1, \hat{y}_2, \ldots, \hat{y}_N\}$ for the sampled points through the following process: for each 3D point $x_n$, it projects the point to all $K$ projection views to obtain 2D coordinates, extracts view-specific feature representations from $\mathcal{E}$ using bilinear interpolation, fuses these multi-view features using the aggregation function $\delta$, and finally decodes the fused representation into an intensity value using the MLP decoder $\psi$. Ground-truth intensities $Y = \{y_1, y_2, \ldots, y_N\}$ are obtained from the CT volume using trilinear interpolation at each $x_n$.

The objective function is the mean squared error between predicted and ground-truth intensities: $\mathcal{L}(Y, \hat{Y}) = \frac{1}{N} \sum_{n=1}^N (y_n - \hat{y}_n)^2$.

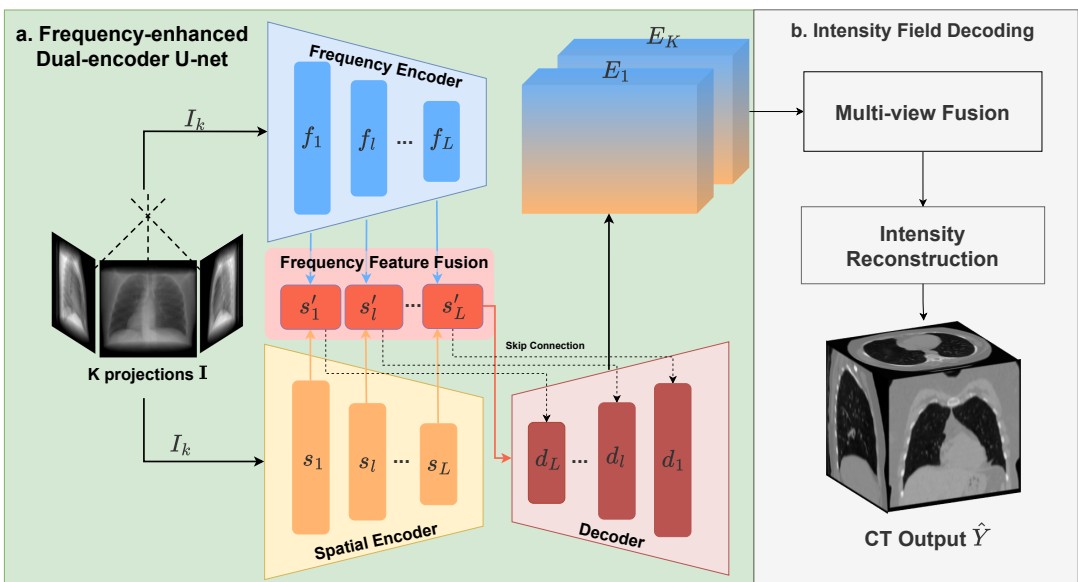

Figure 1: Illustration of DuFal for CT image reconstruction from multiple projection views. Our proposed Frequency-enhanced Dual-encoding (part a) processes a projection $I_k$ from $K$ input projections through two parallel encoders: a Frequency Encoder that analyzes frequency components of $I_k$ to produce frequency features $\{f_1, f_l, \ldots, f_L\}$ and a Spatial Encoder that produces spatial features $\{s_1, s_l, \ldots, s_L\}$. The model then fuses both domains through a CAFF module before decoding through a series of decoder layers at the Feature Decoder with skip connections to produce the feature $E_k$ of the input image $I_k$. By repeating this process for all projections, we obtain the extracted features $\{E_1, \ldots, E_K\}$. The Intensity Field Decoding (part b) operates as in Section 3.2: it first performs Multi-view Fusion by projecting 3D query points to extract view-specific features through bilinear interpolation and then aggregates them using a set function to create unified 3D representations. Subsequently, Intensity Reconstruction employs a multi-layer MLP to decode these fused features into continuous intensity values, which are sampled over a uniform 3D grid to generate the final volumetric CT output $\hat{Y}$.

# 4 Proposed Method: DuFal

The quality of the encoded X-ray features contributes greatly to the quality of the reconstructed CT images. Yet, the encoder, which consists of CNN layers, is inherently biased toward low-frequency information and fails to capture fine details. To this end, we propose **Frequency-enhanced Dual-encoding**. As shown in Figure 1, for a given projection $I_k \in \mathbb{R}^{H_I \times W_I}$, it is processed in parallel by two encoders: a **Frequency Encoder** and a **Spatial Encoder**. The spatial encoder produces a set of $L$ spatial feature maps $\{s_l\}_{l=1}^L$, where $s_l \in \mathbb{R}^{C_l \times H_l \times W_l}$, capturing structural context from the image domain, where $C_l$ is the hidden channel at layer $l$. Simultaneously, the frequency encoder produces a complementary set of high-frequency feature maps $\{f_l\}_{l=1}^L$, where $f_l \in \mathbb{R}^{C_l \times H_l \times W_l}$, enhancing the network's ability to preserve fine anatomical details.

At each resolution layer, the two representations $s_l$ and $f_l$ are fused via **Cross-Attention Frequency Fusion (CAFF)** to obtain a unified feature $s_l' \in \mathbb{R}^{C_l \times H_l \times W_l}$ that integrates both spatial and spectral information. The fused features $\{s_l'\}_{l=1}^L$ are then passed to a **Feature Decoder**, which reconstructs a consolidated feature map $E_k \in \mathbb{R}^{C \times H \times W}$ for each projection $k$. Finally, the fused multi-view features $\mathcal{E} = \{E_1, E_2, \ldots, E_K\}$ are passed through **Intensity Field Decoding** to reconstruct the CT intensity volume $\hat{Y} \in \mathbb{R}^{H_c \times W_c \times D_c}$.

## 4.1 Frequency Encoder

Since Frequency Encoder processes a single projection $I_k$ at a time, we omit the view index $k$ from the image and feature symbols in this section for notational simplicity.

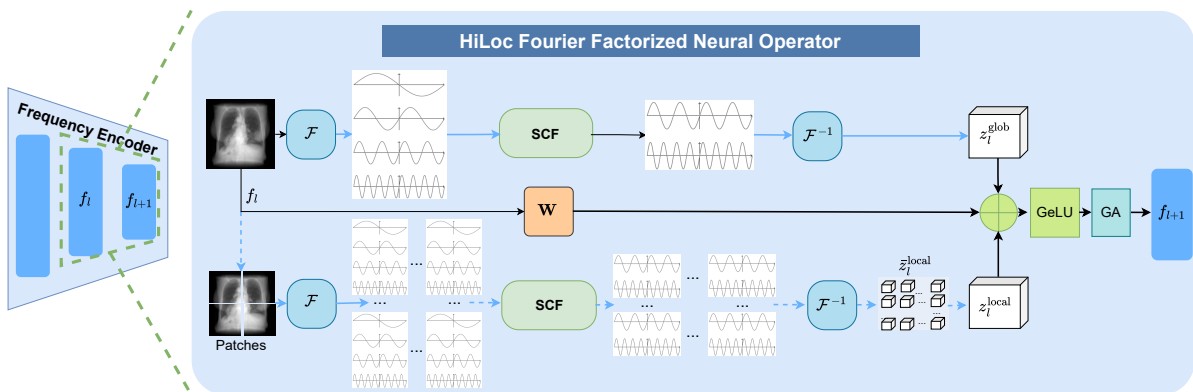

Figure 2: **Overview of the proposed HiLocFFNO block within the Frequency Encoder.** Each HiLocFFNO block comprises two major branches: The gHiF is illustrated at the top using solid blue line (—) and arrows (→), and the lHiF is depicted at the bottom using dashed blue line (- - -) and arrows (--→). Both branches employ SCF to replace the full complex weight with separate channel-mixing and spectral-weighting kernels. After inverse Fourier transform, the resulting frequency features $z_l^{\text{glob}}$ and $z_l^{\text{local}}$ are fused with a linear projection **W**, activated by GeLU, and refined by a Galerkin-Attention (GA) (Cao, 2021) layer to yield the next-level representation $f_{l+1}$.

Our Frequency Encoder is inspired by the FNO architecture (Li et al., 2020) (Kalimuthu et al., 2025). Despite the advances of FNO in capturing frequency domain features, which help them excel in solving partial differential equation tasks dominated by low-frequency components, they exhibit limitations in natural and medical imaging where high-frequency features are equally essential, such as edges and abnormalities. To address this challenge, we propose a novel FNO architecture that enhances high-frequency information capture at both the local detail level and the global context level. Particularly, our Frequency Encoder comprises multiple FNO-based modules, collectively referred to as HiLocFFNO(·), forming an iterative architecture:

$$I \xmapsto{\text{HiLocFFNO}} f_1 \xmapsto{\text{HiLocFFNO}} f_2 \xmapsto{\text{HiLocFFNO}} \cdots \xmapsto{\text{HiLocFFNO}} f_L, \tag{2}$$

where each $f_l \in \mathbb{R}^{C_l \times H_l \times W_l}$ for $l = 1, 2, \ldots, L$, $f_{l+1} = \text{HiLocFFNO}(f_l)$, and $f_1 = \text{HiLocFFNO}(I)$. While $I$ is a grayscale image, we can view $I$ as a feature map with shape $1 \times H_I \times W_I$. Each HiLocFFNO(·) contains two branches: a **Global High-frequency Enhanced FNO (gHiF)** and a **Local High-frequency Enhanced FNO (lHiF)**. As shown in Figure 2, the gHiF is illustrated at the top using solid arrows (→), while the lHiF is depicted at the bottom using dashed arrows (--→). Let $\mathcal{F}$ denote the Fourier transform and $\mathcal{F}^{-1}$ its inverse, HiLocFFNO is implemented as follows:

**Global High-frequency Enhanced FNO (gHiF).** In our implementation, the input feature map $f_l \in \mathbb{R}^{C_l \times H_l \times W_l}$ is first transformed via a 2D Fourier transform $\mathcal{F}$ applied along the spatial dimensions:

$$z_l = \mathcal{F}(f_l) \in \mathbb{C}^{C_l \times \texttt{modes}_1 \times \texttt{modes}_2}, \tag{3}$$

where $\texttt{modes}_1$ and $\texttt{modes}_2$ denote the number of retained frequency components along the height and width dimensions, respectively. In contrast to vanilla FNO which typically retains low-frequency modes, gHiF **selectively preserves high-frequency components**, enabling better capture of fine-grained features.

In the vanilla FNO, each output coefficient is produced with a full complex weight $R_\phi \in \mathbb{C}^{\bar{C}_l \times C_l \times \texttt{modes}_1 \times \texttt{modes}_2}$, resulting in a total of $2 * \bar{C}_l * C_l * \texttt{modes}_1 * \texttt{modes}_2$ parameters. The output frequency component is given by:

$$\hat{z}_l[\bar{c}_l, u, v] = \sum_{c_l} R_\phi[\bar{c}_l, c_l, u, v] \, z_l[c_l, u, v] \tag{4}$$

where $\bar{c}_l \in \{1, ..., \bar{C}_l\}$ indexes the output channel dimension, $c_l \in \{1, ..., C_l\}$ indexes the input channel dimension, and $(u, v)$ index the frequency coordinates. This can grow rapidly with larger values of $\bar{C}_l$, $\texttt{modes}_1$,

and $\texttt{modes}_2$. In response, inspired by the depthwise separable convolution strategy in MobileNet Howard et al. (2017), we propose SCF that decomposes the weight tensor into two smaller, independent components: $R_{\phi,1} \in \mathbb{C}^{\bar{C}_l \times C_l}$ for channel mixing and $R_{\phi,2} \in \mathbb{C}^{C_l \times \texttt{modes}_1 \times \texttt{modes}_2}$ for spectral weighting. This results in the following computation:

$$\hat{z}_l[\bar{c}_l, u, v] = \sum_{c_l=1}^{C_l} R_{\phi,1}[\bar{c}_l, c_l] \, R_{\phi,2}[c_l, u, v] \, z_l[c_l, u, v]. \tag{5}$$

Our proposed factorized form requires only $2\left(\bar{C}_l * C_l + C_l * \texttt{modes}_1 * \texttt{modes}_2\right)$ parameters, yielding a saving ratio of

$$\frac{2 * \bar{C}_l * C_l + 2 * C_l * \texttt{modes}_1 * \texttt{modes}_2}{2 * \bar{C}_l * C_l * \texttt{modes}_1 * \texttt{modes}_2} = \frac{1}{\texttt{modes}_1 * \texttt{modes}_2} + \frac{1}{\bar{C}_l} \tag{6}$$

relative to the original count. For instance, in the final layer $L^{th}$, given $C_l = 512$, $\bar{C}_l = 1024$, and $\texttt{modes}_1 = \texttt{modes}_2 = 16$, the factorized weight retains only **0.49%** of the original parameters, representing about a **99.51%** reduction.

Finally, an inverse Fourier transform yields the frequency feature

$$z_l^{\text{glob}} = \mathcal{F}^{-1}(\hat{z}_l) \in \mathbb{R}^{\bar{C}_l \times H_l \times W_l}. \tag{7}$$

**Local High-frequency Enhanced FNO (lHiF).** Prioritizing high-frequency modes alone is insufficient to fully capture important fine-grained details. To address this issue, we propose a parallel FNO branch that operates on spatially partitioned patches while retaining all frequency modes to capture local information. Using the same input feature $f_l$ as the global branch, we partition it into $N_p \times N_p$ non-overlapping patches, where $N_p = H_l/H_p = W_l/W_p$, resulting in $N_p^2$ local patches $f_{n_p,l} \in \mathbb{R}^{C_l \times H_p \times W_p}$, where $n_p = 0, \dots, N_p^2$. Each patch $f_{n_p,l}$ is independently processed using the same frequency-domain operations with factorized weights (Equations 3, 5, and 7) as in the global branch to produce the local feature $z_{n_p,l}^{\text{local}} \in \mathbb{R}^{\bar{C}_l \times H_p \times W_p}$. We aggregate all patches to obtain the final local feature $z_l^{\text{local}}$ by reshaping the patches from $N_p^2 \times \bar{C}_l \times H_p \times W_p$ back to the full-resolution feature map $z_l^{\text{local}} \in \mathbb{R}^{\bar{C}_l \times H_l \times W_l}$ in the same order as the partitioning step.

Finally, we fuse the outputs of two branches and pass the result to a Galerkin-Attention (GA) (Cao, 2021) layer. After the GA layer, we obtain $f_{l+1}$. Formally, we have

$$f_{l+1} = \text{GA}(\text{GeLU}(\mathbf{W}f_l + z_l^{\text{local}} + z_l^{\text{glob}})), \tag{8}$$

where $\mathbf{W} \in \mathbb{R}^{\bar{C}_l \times C_l}$ is a linear transformation.

## 4.2 Spatial Encoder

At each depth $l$, it performs two unpadded $3 \times 3$ convolutions, each followed by a ReLU, and then applies a $2 \times 2$ max-pooling with stride 2. The max-pool halves the spatial resolution ($H_{l+1} = H_l/2$, $W_{l+1} = W_l/2$), and the subsequent convolutional stage doubles the channel width ($C_{l+1} = 2C_l$). Repeating this pattern for $L$ levels yields the hierarchy of feature maps $\{s_l\}_{l=1}^L$, where each $s_l \in \mathbb{R}^{C_l \times H_l \times W_l}$ is ready for fusion with the frequency features.

## 4.3 Cross-Attention Frequency Fusion (CAFF)

Figure 3 illustrates the overall flow of our CAFF module. Given two input feature maps $s_l, f_l \in \mathbb{R}^{C_l \times H_l \times W_l}$, our goal is to fuse them in the frequency domain. Each feature is first projected via a convolution and transformed into the frequency domain:

$$\hat{s}_l = \mathcal{F}(\text{Conv}(s_l)) = \Re_s + i \cdot \Im_s, \quad \hat{f}_l = \mathcal{F}(\text{Conv}(f_l)) = \Re_f + i \cdot \Im_f, \tag{9}$$

where $\Re_s, \Im_s, \Re_f, \Im_f \in \mathbb{R}^{C_l \times H_l \times W_l}$ are the real and imaginary components, respectively.

To enable spatially-aware feature reweighting, we apply a cross-attention operator $\text{CA}(k, v, q)$ separately to the real and imaginary parts, where $k$, $v$, and $q$ denote the key, value, and query.

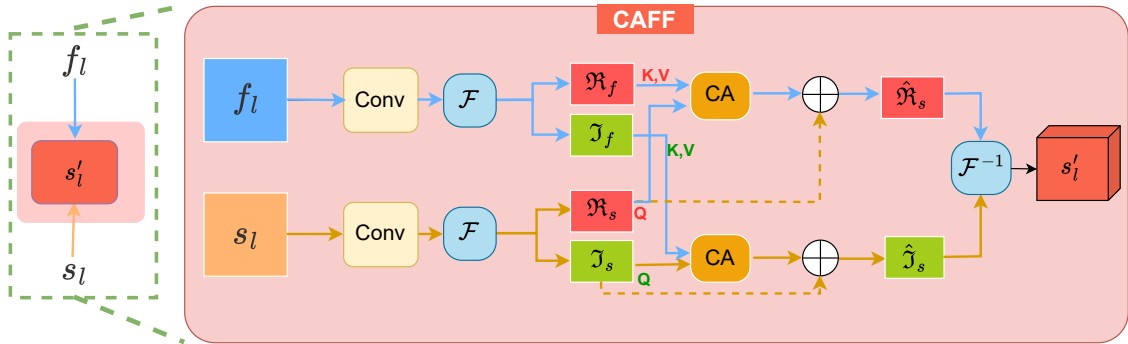

Figure 3: The CAFF flowchart. This diagram shows the fusion process between spatial features $s_l$ and frequency features $f_l$ in the frequency domain. Both input features are first convolved and transformed into the frequency domain using Fourier transform $\mathcal{F}$, decomposing into real ($\Re_f$, $\Re_s$) and imaginary ($\Im_f$, $\Im_s$) components. Cross-attention (CA) operations are applied separately to real and imaginary parts, where frequency components serve as keys/values and spatial components as queries. The attended frequency components are combined with original spatial frequency components ($\Re_s$, $\Im_s$) through residual addition $\oplus$, then transformed back to spatial domain via inverse Fourier transform $\mathcal{F}^{-1}$ to produce the fused feature $s_l'$. → indicates the frequency processing path, → shows the spatial processing path, and ⇢ represents the residual connections.

$$\bar{\Re}_f = \mathrm{CA}(\Re_f, \Re_f, \Re_s), \quad \bar{\Im}_f = \mathrm{CA}(\Im_f, \Im_f, \Im_s). \tag{10}$$

The attended frequency components ($\bar{\Re}_f$ and $\bar{\Im}_f$) are then combined with the original spatial features via residual addition:

$$\hat{\Re}_s = (\Re_s + \bar{\Re}_s), \quad \hat{\Im}_s = (\Im_s + \bar{\Im}_s). \tag{11}$$

Finally, the fused complex-valued representation is transformed back into the spatial domain via an inverse Fourier transform:

$$s_l' = \mathcal{F}^{-1}(\hat{\Re}_s + i\hat{\Im}_s) \in \mathbb{R}^{C_l \times H_l \times W_l}. \tag{12}$$

### 4.4 Feature Decoder

Starting from the deepest fused feature $s_L'$, it performs $L$ upsampling stages. At stage $l$ ($l = L, \ldots, 2$), a $2 \times 2$ transposed convolution doubles the spatial size, concatenates the result with the skip connection $s_{l-1}'$, and refines the concatenated tensor using two successive $3 \times 3$ Conv-ReLU blocks $d_{l-1} = \mathrm{CAT}(\mathrm{Up}(d_l), s_{l-1}')$, where $d_L = s_L'$, $\mathrm{Up}(\cdot)$ denotes transposed convolution, and $\mathrm{CAT}(\cdot, \cdot)$ denotes the two-block concatenation. This sequence yields decoder features $\{d_l\}_{l=1}^L$ with $d_l \in \mathbb{R}^{C_l \times H_l \times W_l}$. Finally, a $1 \times 1$ convolution projects the top-level feature to the projection representation $E_k = \mathrm{Conv}_{1 \times 1}(d_1) \in \mathbb{R}^{C \times H \times W}$. At the end of our Fourier-augmented Dual-encoding, we obtain the encoded features of $K$ projections $\mathcal{E} = \{E_1, E_2, \ldots, E_K\}$, which are then decoded via Intensity Field Decoding described in Section 3.2.

## 5 Experiments

### 5.1 Experiment setup

**Datasets.** We evaluate our method on two CT datasets. The **LUNA16 dataset** (Setio et al., 2017) contains 888 chest CT scans split into 738 training, 50 validation, and 100 testing samples. Scans are resampled and standardized to $256^3$ voxels with $[1.6, 1.6, 1.6]$ mm spacing, following Lin et al. (2024c). The **ToothFairy dataset** (Cipriano et al., 2022) consists of 443 dental CBCT scans with varying resolutions split into 343 training, 25 validation, and 75 testing scans. Scans are resampled and standardized to $256^3$ voxels with spacing $[0.54, 0.54, 0.21]$ mm.

Table 2: Performance evaluation on the **LUNA16 dataset**. ↑ indicates that higher values are better. Best results are **bolded** and second-best results are underlined.

| Methods | 6-View | | 8-View | | 10-View | |
|---|---|---|---|---|---|---|
| | PSNR ↑ | SSIM ↑ | PSNR ↑ | SSIM ↑ | PSNR ↑ | SSIM ↑ |
| FDK (Feldkamp et al., 1984) | 15.36 | 31.41 | 15.95 | 31.00 | 16.25 | 31.79 |
| SART (Andersen & Kak, 1984) | 18.94 | 49.47 | 20.60 | 54.63 | 21.75 | 58.94 |
| NAF (Zha et al., 2022a) | 18.76 | 54.16 | 20.51 | 60.84 | 22.17 | 62.22 |
| NeRP (Shen et al., 2022a) | 23.55 | 74.46 | 25.53 | 80.67 | 26.12 | 81.30 |
| Freeseed (Ma et al., 2023) | 25.59 | 77.36 | 26.86 | 78.92 | 27.23 | 79.25 |
| DiF-Net (Lin et al., 2023) | 24.68 | 71.46 | 25.67 | 76.47 | 26.53 | 76.08 |
| DiF-Gaussian (Lin et al., 2024a) | 26.48 | 81.78 | 26.93 | 82.09 | 29.29 | 87.55 |
| **DuFal (ours)** | **28.20 ± 0.38** | **85.83 ± 0.27** | **28.77 ± 0.42** | **85.50 ± 1.65** | **29.41 ± 0.18** | **87.67 ± 0.22** |

Multi-view 2D projections of size $256 \times 256$ are simulated using an open-source toolbox[1], with viewing angles uniformly sampled over 180°. We experiment with 6, 8, and 10 views to assess the effect of projection count.

**Implementation Details.** We implement the HiLocFFNO as a multi-layer encoder with $L = 4$ iterative blocks. Each block contains a dual-branch FNO module: a global branch preserving the top $modes_1 = modes_2 = 16$ Fourier components and a local branch operating on $16 \times 16 \times 16$ non-overlapping patches. The GA layer fuses global, local, and projected input features using spatial-domain linear projection. The hidden dimension $C$ varies across different stages of the network. The fusion block uses complex-valued cross-attention in the frequency domain with 4 attention heads. All models are trained using the Adam optimizer with a learning rate of $2 \times 10^{-4}$, batch size 4, and a cosine annealing scheduler.

**Baselines.** We compare our model with several existing baselines, including traditional methods FDK (Feldkamp et al., 1984) and SART (Andersen & Kak, 1984), which require no training and are directly inferred using implementations from a public GitHub repository. For NAF (Zha et al., 2022a), NeRP (Shen et al., 2022a), and FreeSeed (Ma et al., 2023), we report the results as provided in (Lin et al., 2024a). Additionally, we reproduce DIF-Net (Lin et al., 2023) and DIF-Gaussian (Lin et al., 2024a) using their official codebases, training for 400 epochs on the LUNA16 dataset and 600 epochs on the ToothFairy dataset following their original paper settings. We note that spatial-domain methods such as C2RV-Net (Lin et al., 2024b) were not publicly available at submission time, precluding comparison on our datasets; however, we provide a comparison using our own implementation in Appendix A.4.

**Evaluation Metrics.** We follow previous works (Zha et al., 2022a; Lin et al., 2023) to evaluate the reconstructed CT using the following metrics: Peak Signal-to-Noise Ratio (PSNR) in decibels (dB) and Structural Similarity Index Measurement (SSIM) in percentage (%). Higher PSNR/SSIM values indicate better reconstruction quality. We also report weighted scores (W-PSNR/W-SSIM) in the ablation study to provide an additional perspective on the metrics when focusing only on regions of interest.

## 5.2 Quantitative Results

**Results on LUNA16 Dataset.** As shown in Table 2, DuFal consistently achieves the highest average metrics across all view settings (6, 8, and 10 views). This advantage is particularly pronounced in very sparse view configurations such as 6 and 8 views, indicating that our enhanced frequency feature capture capability provides proven benefits even in ill-posed sparse view settings. Among neural methods, NeRP typically underperforms compared to more recent approaches, while DIF-Gaussian ranks as the second-best method in most settings, though our approach consistently surpasses both across all view configurations.

---

[1] https://github.com/CERN/TIGRE

Table 3: Performance evaluation on the **ToothFairy dataset**. ↑ indicates that higher values are better. Best results are **bolded** and second-best results are underlined.

| Methods | 6-View | | 8-View | | 10-View | |
|---|---|---|---|---|---|---|
| | **PSNR ↑** | **SSIM ↑** | **PSNR ↑** | **SSIM ↑** | **PSNR ↑** | **SSIM ↑** |
| FDK (Feldkamp et al., 1984) | 17.07 | 39.90 | 18.42 | 43.29 | 19.58 | 47.21 |
| SART (Andersen & Kak, 1984) | 20.04 | 64.98 | 21.92 | 67.86 | 22.82 | 71.53 |
| NAF (Zha et al., 2022a) | 20.58 | 63.52 | 22.39 | 67.24 | 23.84 | 72.52 |
| NeRP (Shen et al., 2022a) | 21.27 | 72.06 | 24.18 | 78.83 | 25.99 | 82.08 |
| FreeSeed (Ma et al., 2023) | 26.35 | 78.98 | 27.08 | 81.38 | 27.63 | 84.40 |
| DiF-Net (Lin et al., 2023) | 25.55 | 84.40 | 26.09 | 85.07 | 26.67 | 86.09 |
| DiF-Gaussian (Lin et al., 2024a) | 25.74 | 74.64 | 26.27 | 75.23 | 28.60 | 83.11 |
| **DuFal (ours)** | **28.79 ± 0.27** | **86.08 ± 1.20** | **29.45 ± 0.58** | **87.56 ± 1.14** | **30.51 ± 0.47** | **89.49 ± 0.51** |

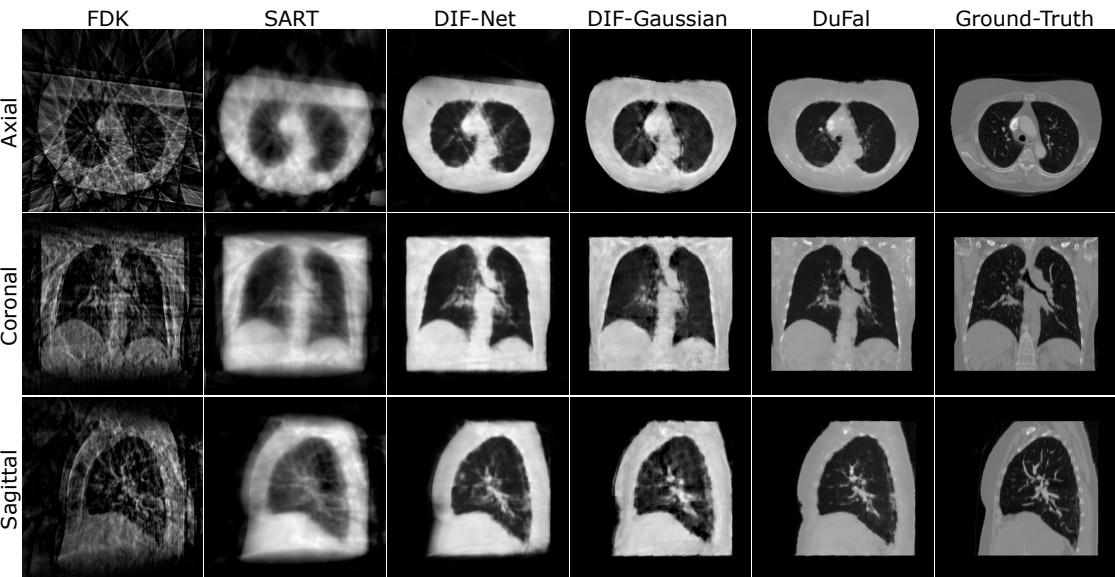

Figure 4: Visualization of 10-view reconstructed chest CT (from top to bottom: axial, coronal, and sagittal slice). This figure highlights the key advantage of our method in preserving fine details. While DIF-Net and DIF-Gaussian successfully recover the general anatomy, they produce overly smoothed images that lose high-frequency information, as evidenced by the blurry details in the zoomed-in views. In contrast, our method yields visibly sharper reconstructions across the axial, coronal, and sagittal planes, achieving a fidelity much closer to the ground-truth image.

**Results on ToothFairy Dataset.** As reported in Table 3, our method consistently achieves the highest average metrics across all view settings (6, 8, and 10 views), with SSIM showing notable improvements over baseline methods. Similar to the LUNA16 dataset, our model demonstrates enhanced performance, particularly in very sparse view configurations such as 6 and 8 views, confirming its robustness under limited projection scenarios. FreeSeed, a post-processing model that emphasizes the ability to capture high-frequency details by incorporating FFT layers, outperforms DIF-Gaussian in sparse view settings like 6 or 8 views but generally underperforms compared to our approach across all view configurations. Among other INR-based methods, NeRP and DIF-Net typically underperform compared to more recent approaches, while our method consistently surpasses all baselines, demonstrating the effectiveness of our enhanced frequency feature capture capability for dental CT reconstruction.

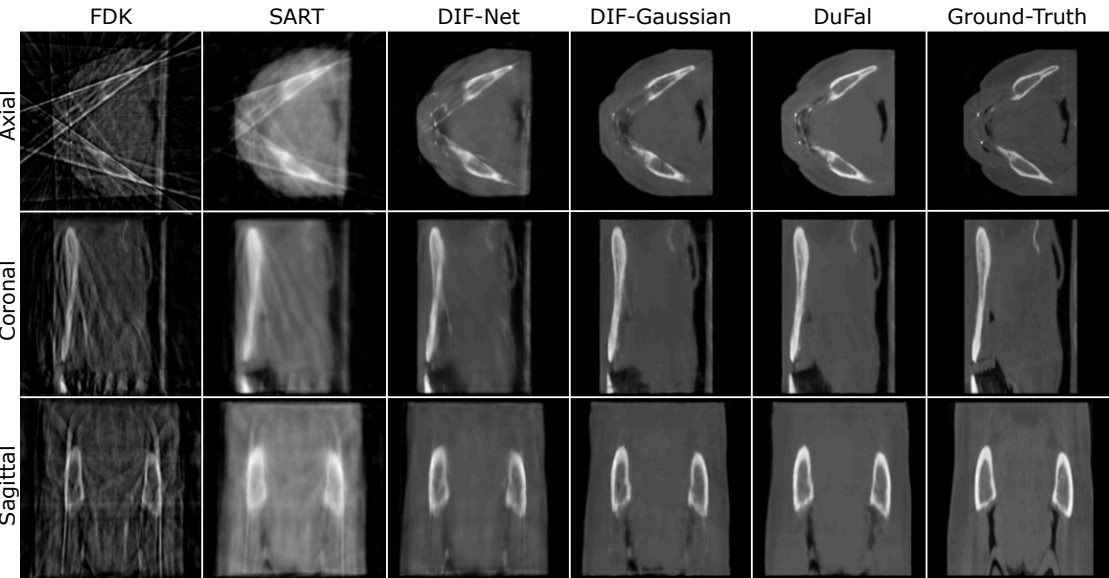

Figure 5: Visualization of 10-view reconstructed dental CT (from top to bottom: axial, coronal, and sagittal slice). While the competing DIF-Gaussian method reconstructs the general bone structure, it oversmooths the image, completely obscuring fine details like the mandibular canals. Our method yields a significantly sharper result that clearly defines the cortical bone and successfully resolves the mandibular canals in the coronal and sagittal views.

### 5.3 Qualitative Analysis

We present visual comparisons to further demonstrate the differences between our model and baseline methods. We focus on 6-view reconstructions as this represents the most challenging sparse-view scenario and best showcases reconstruction quality differences (Figures 4 and 5). Traditional methods such as FDK(Feldkamp et al., 1984) and SART (Andersen & Kak, 1984) exhibit noticeable artifacts and fail to preserve fine anatomical structures, while DIF-Net (Lin et al., 2023) and DIF-Gaussian (Lin et al., 2024a) show improvements but still struggle to capture critical high-frequency details and tissue boundaries. Our model produces reconstructions with enhanced clarity and accurate contrast representation, effectively preserving high-frequency features while minimizing artifacts. On the ToothFairy dataset, similar patterns emerge: traditional methods fail to recover fine dental structures. Our approach demonstrates improved capability in reconstructing sharp boundaries and complex anatomical features while maintaining high fidelity under sparse projection conditions across different medical imaging domains.

## 6 Ablation Study

### 6.1 The Effect of lHiF in HiLocFFNO

Table 4 evaluates the contribution of the lHiF branch within the HiLocFFNO block. This component captures local spatial relationships through patch-based processing, complementing the global frequency analysis in gHiF. The results demonstrate that including the lHiF branch provides performance improvements across both datasets, with particularly notable gains on the ToothFairy dataset. This validates the importance of combining both gHiF and lHiF for optimal reconstruction quality.

### 6.2 Comparison between CAFF and Naive Cross-Attention

Table 5 presents a comparative analysis between our proposed CAFF and conventional spatial-domain cross-attention mechanisms. The baseline spatial cross-attention approach processes both spatial feature

Table 4: Ablation study for lHiF in HiLocFFNO.

| lHiF | LUNA16 | | ToothFairy | |
|---|---|---|---|---|
| | PSNR ↑ | SSIM ↑ | PSNR ↑ | SSIM ↑ |
| ✗ | 29.56 | 87.14 | 29.45 | 86.02 |
| ✓ | **29.63** | **87.91** | **30.79** | **89.67** |

$s_l$ and frequency feature $f_l$ through convolution operations before applying cross-attention in the spatial domain, where $s_l$ serves as queries and $f_l$ functions as keys and values. In contrast, our CAFF operates cross attention on the frequency domain to integrate multi-modal feature representations. The experimental results demonstrate that CAFF achieves superior reconstruction performance compared to spatial cross-attention (CA), confirming the effectiveness of frequency-domain feature fusion for sparse-view CBCT reconstruction.

Table 5: Ablation study comparing different feature fusion strategies on the LUNA16 and ToothFairy datasets.

| Fusion Method | LUNA16 | | ToothFairy | |
|---|---|---|---|---|
| | PSNR ↑ | SSIM ↑ | PSNR ↑ | SSIM ↑ |
| Add | 29.28 | 87.40 | 29.77 | 88.52 |
| Concat | 29.21 | 87.33 | 29.70 | 88.43 |
| Spatial CA | 29.59 | 87.25 | 30.75 | 89.49 |
| CAFF (ours) | **29.63** | **87.91** | **30.79** | **89.67** |

Table 6: Impact of Factorization on PSNR, SSIM, and Model Parameters.

| Factorization | LUNA16 | | ToothFairy | | Model Size | |
|---|---|---|---|---|---|---|
| | PSNR | SSIM | PSNR | SSIM | Params (M) | Reduction |
| No Factorization | 29.54 | 87.94 | 30.94 | 90.27 | 445M | - |
| Factorized | 29.63 (+0.09) | 87.91 (-0.03) | 30.79 (-0.15) | 89.67 (-0.6) | 79M | 82.2% |

## 6.3 SCF's Factorization Analysis

Table 6 evaluates the impact of factorization in SCF on performance and computational efficiency across both datasets. Importantly, the factorized variant maintains comparable reconstruction quality, with only minimal variations in PSNR and slight decreases in SSIM scores, demonstrating that reconstruction quality is largely preserved despite the significant parameter reduction. Most significantly, factorization achieves a substantial 82.2% reduction in model parameters, decreasing from 445M to 79M parameters. This dramatic computational efficiency gain demonstrates that factorization maintains comparable performance while substantially reducing parameter complexity, making the approach more practical for deployment in resource-constrained environments.

## 6.4 HiLocFFNO Plug-and-Play Integration

Table 7 presents an ablation study on the plug-and-play integration of the proposed HiLocFFNO into various reconstruction models across the LUNA16 and ToothFairy datasets. The inclusion of FFNO consistently improves performance, with particularly notable gains on the DIF-Gaussian model, where PSNR and SSIM show improvements on both datasets. This suggests that FFNO effectively enhances both image fidelity and structural quality.

## 6.5 Efficiency Analysis

As shown in Table 8 and Figure 6, DuFal demonstrates a favorable computational trade-off, particularly in inference speed. Most notably, our method achieves significantly faster inference than DiF-Net (1.938 vs 3.778 s/sample), and the training time of DuFal does not scale directly with parameter count. In medical AI applications where precision is the primary concern, this computational trade-off is highly acceptable given the quality improvements.

While DuFal's training time is longer due to the dual-path architecture and Fourier operator computations, this overhead is incurred only once during offline training. The model's faster inference speed provides clear advantages in real-world clinical deployment. Furthermore, training efficiency can be substantially improved through distributed data-parallel training with multiple GPUs and larger batch sizes. Future work could explore initialization from pre-trained medical foundation models to accelerate convergence while maintaining robust performance.

Table 7: Ablation study of plug-and-play of the HiLocFFNO modular on different models.

| Methods | HiLocFFNO | LUNA16 | | ToothFairy | |
|---|---|---|---|---|---|
| | | PSNR ↑ | SSIM ↑ | PSNR ↑ | SSIM ↑ |
| FBPConvNet (Lin et al., 2023) | ✗ | 25.35 | 75.01 | 27.01 | 76.12 |
| | ✓ | **25.55** | **75.68** | **27.35** | **76.86** |
| DiF-Net (Lin et al., 2023) | ✗ | 26.53 | 76.08 | 26.67 | 86.09 |
| | ✓ | **26.85** | **76.62** | **27.79** | **86.49** |
| DiF-Gaussian (Lin et al., 2024a) | ✗ | 29.29 | 87.55 | 28.60 | 83.11 |
| | ✓ | **29.63** | **87.91** | **30.79** | **89.67** |

Table 8: Computational efficiency comparison on the LUNA16 dataset. ↓ indicates that lower values are better for speed metrics.

| Methods | Params (M) ↓ | Inference Speed (s/sample) ↓ | Training Time (h) ↓ |
|---|---|---|---|
| Freeseed | 48.41 | 1.274 | 29.6 |
| DiF-Net | 31.11 | 3.778 | 24.5 |
| DiF-Gaussian | 31.71 | 0.695 | 26.5 |
| **DuFal (ours)** | 78.92 | 1.938 | 34.1 |

## 6.6 Ablation on High-Fidelity Reconstruction

### 6.6.1 ROI-Weighted Metrics.

Conventional PSNR and SSIM can be misleading when degradation occurs in clinically important regions. As shown in Figure 7, adding Gaussian noise or Laplace noise only inside the lung ROI (c) yields a higher PSNR (10.59 and 12.12 dB) than adding the same noise to the background (d, 10.37 and 11.78 dB), even though radiologists prefer (d) over (c). To address this issue, we introduce ROI-weighted metrics (W-PSNR, W-SSIM), which focus evaluation on diagnostically important regions. For example, W-PSNR assigns a score of 38.15 dB to the image after background masking, correctly reflecting its perceptual quality.

Under the ROI-weighted metrics, Tables 9 and 10 demonstrate that our method also achieves the highest W-PSNR and W-SSIM across all sparse-view settings on both LUNA16 and ToothFairy datasets. For example, on LUNA16, DuFal achieves 28.52 dB in W-PSNR and 88.96 in W-SSIM, outperforming the runner-up method DiF-Gaussian by +1.15 dB and +2.05 points, respectively. Similar improvements are observed on the ToothFairy dataset (Table 10), where our method consistently surpasses DiF-Gaussian across all sparse-view

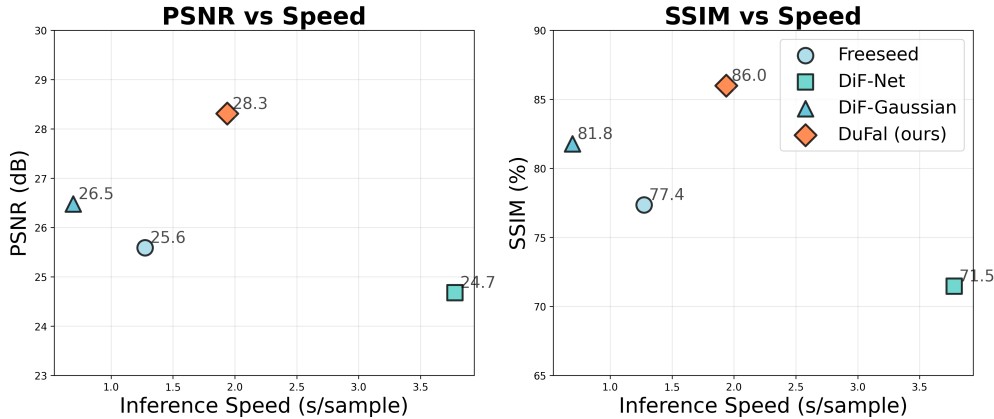

Figure 6: **Performance vs. Inference Speed Trade-off on LUNA16 Dataset (6-View).** Scatter plots comparing reconstruction quality against computational efficiency for different methods. **Left:** PSNR vs. Inference speed. **Right:** SSIM vs. Inference speed. DuFal achieves the highest reconstruction quality in both metrics while maintaining good inference speed.

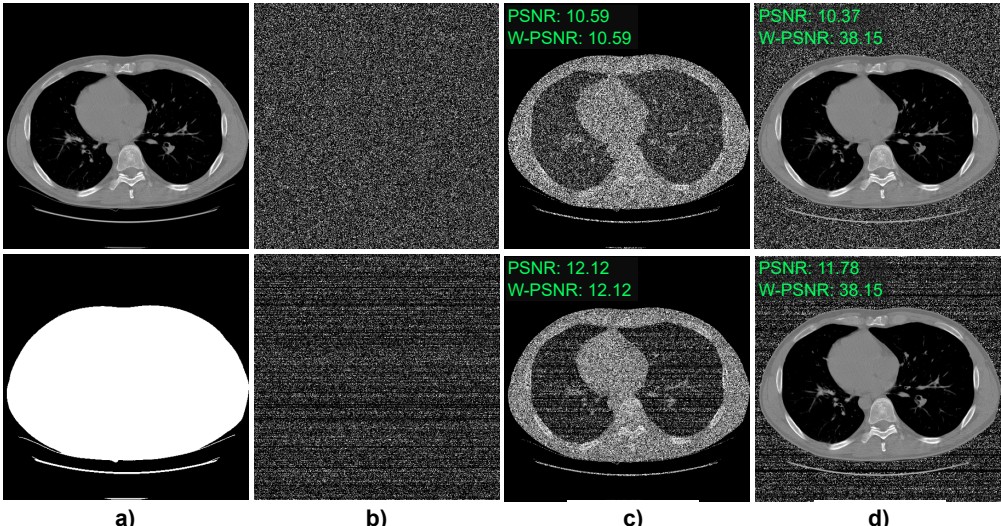

Figure 7: (a) Original image and region of interest in mask format; (b) Two types of noise: Gaussian noise (top) and Laplace noise (bottom); (c) Noise added only within the lung region of interest; (d) Noise added only to the background. Global PSNR favors (c), whereas ROI-weighted PSNR (W-PSNR) correctly ranks (d) higher, demonstrating the need for ROI-weighted metrics.

configurations. The consistent superiority across both datasets under conventional and ROI-weighted metrics establishes the robustness of our proposed approach.

### 6.6.2 Downstream task: Segmentation

To evaluate the clinical utility of our reconstructed CT volumes, we conduct downstream segmentation tasks that assess whether the preserved anatomical details enable accurate automated analysis. High-fidelity reconstruction is crucial for the segmentation task because segmentation algorithms rely on clear anatomical boundaries and structural details that can be compromised by reconstruction artifacts or blurring.

For lung segmentation evaluation on LUNA16, we employed the pre-trained LungMask model (Hofmanninger et al., 2020) for automated lung region extraction. Traditional CBCT reconstruction methods (FDK and

Table 9: Performance ROI-weighted evaluation on the LUNA16 dataset.

| Methods | 6-View | | 8-View | | 10-View | |
|---|---|---|---|---|---|---|
| | W-PSNR ↑ | W-SSIM ↑ | W-PSNR ↑ | W-SSIM ↑ | W-PSNR ↑ | W-SSIM ↑ |
| FDK (Feldkamp et al., 1984) | 15.82 | 72.68 | 16.51 | 74.21 | 16.83 | 75.32 |
| SART (Andersen & Kak, 1984) | 21.21 | 81.43 | 22.57 | 82.37 | 23.52 | 83.32 |
| DiF-Net (Lin et al., 2023) | 26.16 | 85.59 | 26.69 | 86.51 | 27.66 | 87.19 |
| DiF-Gaussian (Lin et al., 2024a) | 27.37 | 86.91 | 27.84 | 87.26 | 30.06 | 90.60 |
| **DuFal (ours)** | **28.52** | **88.96** | **29.83** | **90.29** | **30.59** | **91.03** |

Table 10: Performance ROI-weighted evaluation on the ToothFairy dataset.

| Methods | 6-View | | 8-View | | 10-View | |
|---|---|---|---|---|---|---|
| | W-PSNR ↑ | W-SSIM ↑ | W-PSNR ↑ | W-SSIM ↑ | W-PSNR ↑ | W-SSIM ↑ |
| DiF-Gaussian (Lin et al., 2024a) | 26.29 | 80.73 | 27.47 | 83.11 | 28.08 | 83.97 |
| **DuFal (ours)** | **27.11** | **81.90** | **27.92** | **83.35** | **28.52** | **84.62** |

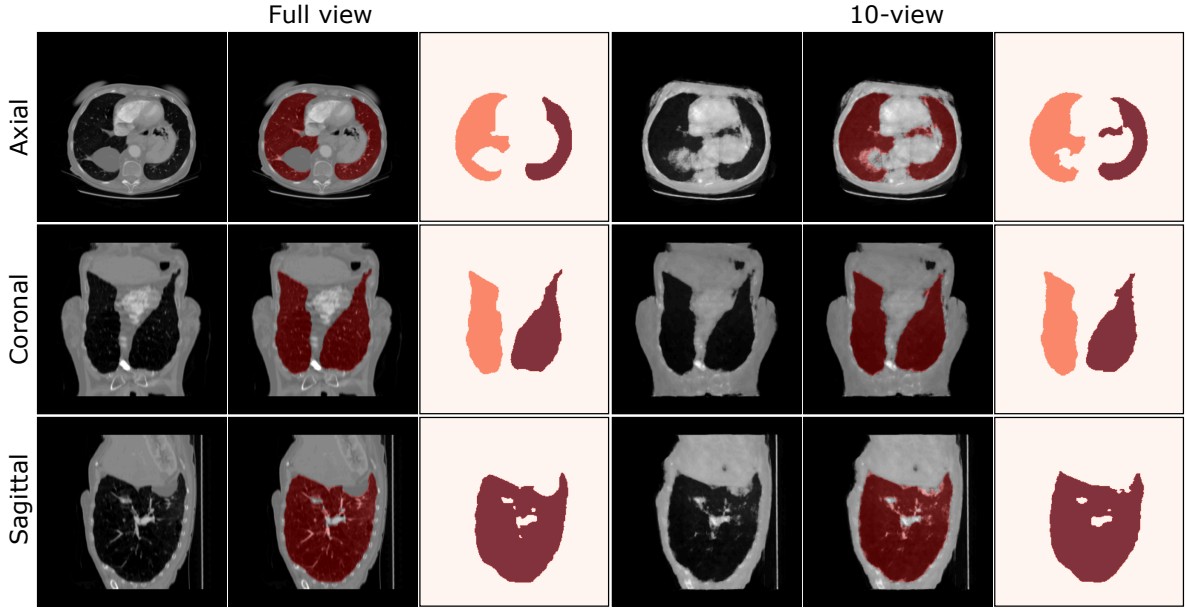

Figure 8: Comparison of LungMask segmentation performance on full-view versus 10-view reconstructed CT images across three anatomical planes. Left panel shows results on full-view (ground truth) CTs, right panel shows results on 10-view reconstructed CTs. For each view (axial, coronal, sagittal from top to bottom), three columns display: (1) original CT slice, (2) segmentation mask overlaid on CT slice in red, and (3) isolated predicted lung mask.

SART) exhibit poor segmentation performance, with Dice scores shown in Table 11, indicating severe degradation of anatomical boundaries in extremely sparse-view scenarios. In contrast, deep learning-based reconstruction methods achieve substantially better segmentation performance, with Dice scores above 96. Figure 8 illustrates the qualitative segmentation results and further confirms DuFal's strong performance. DuFal demonstrates robust preservation of anatomical boundaries with the highest precision and recall values, achieving the best overall segmentation performance.

Table 11: Lung Segmentation Comparison on LUNA16 Dataset.

| Model | Dice↑ | Jaccard↑ | Precision↑ | Recall↑ |
|---|---|---|---|---|
| Full-Views | 97.73 | 95.60 | 97.10 | 98.37 |
| FDK | 8.89 | 4.82 | 6.96 | 12.31 |
| SART | 54.45 | 39.22 | 58.94 | 50.99 |
| DIF-Net | 96.68 | 92.12 | 95.41 | 97.98 |
| **DuFal (ours)** | 97.05 | 94.29 | 95.69 | 98.45 |

## 7 Conclusion

This work presents the DuFal framework, a novel approach to address the extremely sparse-view CBCT reconstruction challenge. Our proposed framework fundamentally addresses the limitations of conventional CNN-based methods that struggle to capture the high-frequency anatomical details essential for accurate medical imaging. Through the introduction of a Frequency-Enhanced Dual-encoding architecture, we demonstrate that simultaneous processing of spatial structural context and frequency domain characteristics significantly improves reconstruction quality in sparse X-ray projection scenarios. The key innovation lies in our HiLocFFNO blocks, which effectively combine global high-frequency enhancement (gHiF) with local patch-based processing (lHiF), ensuring that both major anatomical structures (such as organ boundaries and bone structures) and fine-grained details such as pulmonary nodules are preserved. Our SCF technique achieves remarkable parameter efficiency while maintaining reconstruction quality. Additionally, our CAFF mechanism enables effective integration of spatial and spectral features in the frequency domain. Therefore, DuFal sets a new standard for sparse-view CBCT reconstruction that prioritizes both computational efficiency and medical imaging reconstruction performance.

**Discussion & Future Work.**

- **Modality generalization:** The proposed DuFal has been extentively evaluated on CT datasets, demonstrating its effectiveness in preserving high-frequency anatomical details under extremely sparse-view conditions. While our current experiments focus on CT, many other medical imaging modalities, such as MRI, PET, and ultrasound could potentially benefit from this approach, offering a promising avenue for future research.

- **Enhanced interpretability:** While many prior approaches implicitly learn frequency information, DuFal explicitly models it, making the learned representations more amenable to interpretation. Future work could extend this concept by disentangling additional semantic or structural features, potentially enabling explanations that align more closely with clinical reasoning.

- **Artifact mitigation:** DuFal's frequency-spatial dual architecture offers promising potential for handling motion artifacts. The key insight is that motion artifacts affect spatial and frequency representations differently, creating detectable inconsistencies between domains, whereas anatomical features should exhibit consistent patterns across both representations. However, our current work focuses exclusively on static sparse-view reconstruction, as publicly available datasets with well-annotated motion artifacts are lacking. As future work, we plan to collaborate with clinical partners to evaluate DuFal's robustness on real dynamic CBCT datasets.

**Broader Impact.** The DuFal framework represents a significant advancement in sparse-view CBCT reconstruction with direct clinical implications for routine screening, emergency imaging, and pediatric applications where radiation reduction is critical. The integration of frequency-aware processing addresses the fundamental challenge of preserving high-frequency anatomical details crucial for accurate diagnosis, while our modular design enables integration into existing deep learning pipelines.

**Acknowledgments**   This material is based upon work supported by the National Science Foundation (NSF) under Award No NSF 2443877, 1946391, 2223793 EFRI BRAID, National Institutes of Health (NIH) 1R01CA277739-01. The authors thank the International Max Planck Research School for Intelligent Systems (IMPRS-IS) for supporting Duy M. H. Nguyen.

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

# A    Appendix

## A.1    Analysis of Query-Key-Value Configuration in CAFF

To examine the rationale behind our cross-domain attention design, we conduct an additional ablation study that interchanges the roles of spatial and frequency features in the attention mechanism. As shown in Table 12, reversing the roles results in a consistent performance drop across both datasets. This outcome supports our design choice of assigning spatial features as queries and frequency features as keys and values. Spatial features capture the primary structural context of anatomical regions, while frequency features encode complementary high-frequency details. Allowing spatial tokens to query frequency representations enables more coherent integration between global frequency cues and localized spatial patterns. Conversely, when frequency tokens query spatial ones, the model loses spatial anchoring, leading to less effective feature fusion and degraded reconstruction quality.

Table 12: Ablation study on query-key-value domain configuration in the CAFF module.

| Q | K-V | LUNA16 | | ToothFairy | |
|---|---|---|---|---|---|
| | | PSNR ↑ | SSIM ↑ | PSNR ↑ | SSIM ↑ |
| Frequency | Spatial | 29.24 | 87.32 | 29.67 | 88.41 |
| Spatial | Frequency | **29.63** | **87.91** | **30.79** | **89.67** |

## A.2    Sensitivity Analysis of Frequency Modes

To assess the sensitivity of the global High-Frequency (gHiF) branch to the number of retained Fourier modes, we conducted experiments by halving and doubling the default number of modes (16). Table 13 shows the performance on both datasets with 10 views.

Table 13: Sensitivity analysis for frequency modes in gHiF (10 views).

| Modes | LUNA16 | | ToothFairy | |
|---|---|---|---|---|
| (gHiF) | PSNR ↑ | SSIM ↑ | PSNR ↑ | SSIM ↑ |
| 8 | 29.26 | 87.23 | 30.11 | 88.97 |
| 16 | **29.63** | **87.91** | **30.79** | **89.67** |
| 32 | 29.22 | 87.35 | 30.15 | 89.05 |
| Mean/Std | 29.37 / 0.03 | 87.50 / 0.09 | 30.35 / 0.10 | 89.23 / 0.10 |

The results exhibit low standard deviations ($< 0.1$) across both metrics and datasets, indicating that DuFal maintains stable reconstruction quality despite variations in the number of frequency modes. Increasing and decreasing the number of modes slightly degrade reconstruction quality, confirming that the chosen configuration achieves a good balance between capturing high-frequency components and maintaining stable optimization. These findings further demonstrate that DuFal is robust to moderate changes in the frequency resolution of the gHiF branch, validating the general stability of our frequency modeling strategy.

## A.3    Impact of Local Patch Size

We evaluate the influence of patch size in the local High-Frequency (lHiF) branch on preserving fine spatial structures. Table 14 compares different non-overlapping patch sizes ($8^3$, $16^3$, $32^3$).

DuFal demonstrates stable performance with low standard deviations across all patch sizes, confirming that the model is robust to changes in local partitioning. Both smaller ($8^3$) and larger ($32^3$) patch sizes lead to only

Table 14: Ablation study on local patch size in lHiF.

| Patch Size | LUNA16 | | ToothFairy | |
|---|---|---|---|---|
| (lHiF) | PSNR ↑ | SSIM ↑ | PSNR ↑ | SSIM ↑ |
| $8^3$ | 29.31 | 87.58 | 29.70 | 88.40 |
| $16^3$ | **29.63** | **87.91** | **30.79** | **89.67** |
| $32^3$ | 29.26 | 87.35 | 29.71 | 88.36 |
| Mean/Std | 29.4 / 0.03 | 87.61 / 0.05 | 30.1 / 0.26 | 88.81 / 0.37 |

Table 15: Comparison with C2RV-Net under different initialization settings on the LUNA16 dataset.

| Methods | Init. | Scale | 6-View | | 8-View | | 10-View | |
|---|---|---|---|---|---|---|---|---|
| | | | PSNR↑ | SSIM↑ | PSNR↑ | SSIM↑ | PSNR↑ | SSIM↑ |
| FDK | Scratch | Single | 15.36 | 31.41 | 15.95 | 31.00 | 16.25 | 31.79 |
| SART | Scratch | Single | 18.94 | 49.47 | 20.60 | 54.63 | 21.75 | 58.94 |
| NAF | Scratch | Single | 18.76 | 54.16 | 20.51 | 60.84 | 22.17 | 62.22 |
| NeRP | Scratch | Single | 23.55 | 74.46 | 25.53 | 80.67 | 26.12 | 81.30 |
| Freeseed | Scratch | Single | 25.59 | 77.36 | 26.86 | 78.92 | 27.23 | 79.25 |
| DiF-Net | Scratch | Single | 24.68 | 71.46 | 25.67 | 76.47 | 26.53 | 76.08 |
| DiF-Gaussian | Scratch | Single | 26.48 | 81.78 | 26.93 | 82.09 | 29.29 | 87.55 |
| DiF-Gaussian | Pre-trained | Single | 27.50 | 84.73 | 28.48 | 85.69 | 29.65 | 88.70 |
| C2RV-Net | – | Multiple | 29.23 | 87.47 | 29.95 | 88.46 | 30.70 | 89.16 |
| **DuFal** (Ours) | Scratch | Single | 28.31 | 86.00 | 29.13 | 87.41 | 29.63 | 87.91 |
| **DuFal** (Ours) | Pre-trained | Single | **29.58** | **88.14** | **30.74** | **89.61** | **31.49** | **90.59** |

slight performance degradation compared to the default configuration. Smaller patches may overemphasize fine texture variations, while larger patches reduce spatial locality and weaken boundary precision. The default ($16^3$) configuration achieves the best overall balance between local detail preservation and contextual representation, as reflected by the highest mean PSNR/SSIM. These findings validate that DuFal maintains consistent reconstruction quality even under varying local frequency resolutions.

## A.4 Comparison with C2RV-Net

We further compare DuFal against the recent C2RV-Net Lin et al. (2024c) (CVPR 2024). It is important to note that DuFal is a modular frequency enhancement framework rather than a standalone architecture; its frequency-aware components can be integrated into existing spatial-domain reconstruction networks to improve high-frequency recovery. To validate this property, we implemented DuFal based on the spatial DiF-Gaussian model using pre-trained ResNet-50 Nguyen et al. (2023). As shown in Table 15, DuFal surpasses both C2RV-Net and DiF-Gaussian under comparable sparse-view configurations on the LUNA16 dataset, demonstrating that the observed improvements primarily stem from our proposed frequency-domain integration.

