# OpenReview forum: "DuFal: Dual-Frequency-Aware Learning for High-Fidelity Extremely Sparse-view CBCT Reconstruction"
_TMLR — Accepted by TMLR_

### Review · Reviewer_S3D9 · 2025-10-13

**Summary Of Contributions:**

The authors propose a novel implicit neural representation (INR) framework for extremely sparse-view cone-beam CT (CBCT) reconstruction. Their approach employs two complementary modules: one operating in the frequency domain and the other in the spatial domain. The outputs from these two branches are merged through a cross-attention-based fusion module that integrates features from both the frequency and spatial domains. The proposed framework demonstrates superior reconstruction performance compared to existing baselines on two widely used datasets. The authors also conduct ablation studies to assess the contributions of individual components of their framework.

**Audience:**

Yes

**Audience Explanation:**

The paper addresses a relevant problem in CBCT reconstruction. Its methodological novelty and demonstrated improvements make it of interest to researchers working on CBCT reconstruction and neural operator methods.

**Broader Impact Concerns:**

I have no concerns regarding the ethical implications of the work.

**Claims And Evidence:**

Yes

**Claims Explanation:**

The paper provides clear empirical evidence through comparisons with baselines and ablation experiments.

**Requested Changes:**

- **All Tables:** Please include the standard deviation of reported metrics and specify over how many model runs or random seeds the averages are computed. This applies to all tables in the paper for better statistical confidence.
- **Page 3, second-to-last line:** The phrase "*2D and map 3D*" is unclear.
- **Page 7:** The proposed **Spectral-Channel Factorization (SCF)** appears conceptually similar to depthwise separable convolution. Are these two operations mathematically equivalent, or is there a key distinction that should be emphasised? I understand that SCF works in the frequency domain while depthwise separable convolution works in the spatial domain but the basic idea seems to be the same and it could be that the two operations are even mathematically equivalent in that if the input to SCF is converted to spatial-domain then the operation that gives the same output as SCF would be a deptpthwise separable conv operation (followed by conversion to frequency domain to match SCF's output). Please elaborate on the similarities and the differences.
- **Page 8:** The paper introduces *Galerkin-Attention* but does not include a reference for it. Please provide a citation.
- **Page 8, Section 4.3 (CAFF):** Does the choice of which branch provides the query (Q) and which provides the key/value (K,V) affect the results?
- **Page 13, Section 6.2:** The differences shown in Table 5 seem small. Are they statistically significant? **Suggestion:** I think an ablation that might better isolate the impact of the cross-attention mechanism itself would be to compare it against a simple concatenation or summation of the two branches’ features without attention, rather than only studying the impact of converting the features to the frequency domain in CAFF. Perhaps the differences would be much larger if the ablation were whether having a module to fuse features is better than no module at all (i.e., simple concatenation or summation).
- **Page 15, Section 6.6.2:** Please clarify whether the segmentation evaluation is performed on a single slice or multiple slices, and if multiple, how those slices are selected.

---

> ### Author Response · Authors · 2025-11-11
> **Response to Reviewer S3D9 - Part 1**
>
> We sincerely thank the reviewer for the detailed and thoughtful feedback, which has helped us clarify and strengthen several aspects of our paper. Below, we provide answers to each of your questions/concerns.
>
> > **A. Including the standard deviation of reported metrics**
>
> We thank the reviewer for this valuable suggestion. To assess the robustness of our results, we conducted additional experiments using **three different random seeds** on both the LUNA16 and ToothFairy datasets. The updated tables (included below) report the mean and standard deviation of PSNR and SSIM metrics across all sparse-view settings. The results show minimal variance across runs, indicating that DuFal produces stable and reproducible performance under different random initializations.
>
> Due to computational and time constraints, we did not repeat all experiments in the paper; however, the consistency observed across these two representative benchmarks provides strong evidence of the reliability and stability of our method. These updated results will be included in the revised manuscript for completeness.
>
> **Table 1.** *Performance evaluation on the LUNA16 dataset. ↑ indicates that higher values are better. Best results are bolded.*
> | ***Methods***      | ***6-View PSNR ↑*** | ***6-View SSIM ↑*** | ***8-View PSNR ↑*** | ***8-View SSIM ↑*** | ***10-View PSNR ↑*** | ***10-View SSIM ↑*** |
> | :----------------- | :-----------------: | :-----------------: | :-----------------: | :-----------------: | :------------------: | :------------------: |
> | FDK                |        15.36        |        31.41        |        15.95        |        31.00        |         16.25        |         31.79        |
> | SART               |        18.94        |        49.47        |        20.60        |        54.63        |         21.75        |         58.94        |
> | NAF                |        18.76        |        54.16        |        20.51        |        60.84        |         22.17        |         62.22        |
> | NeRP               |        23.55        |        74.46        |        25.53        |        80.67        |         26.12        |         81.30        |
> | Freeseed           |        25.59        |        77.36        |        26.86        |        78.92        |         27.23        |         79.25        |
> | DiF-Net            |        24.68        |        71.46        |        25.67        |        76.47        |         26.53        |         76.08        |
> | DiF-Gaussian       |        26.48        |        81.78        |        26.93        |        82.09        |         29.29        |         87.55        |
> | ***DuFal (ours)*** |  ***28.20 ± 0.38*** |  ***85.83 ± 0.27*** |  ***28.77 ± 0.42*** |  ***85.50 ± 1.65*** |  ***29.41 ± 0.18***  |  ***87.67 ± 0.22***  |
>
> **Table 2.** *Performance evaluation on the ToothFairy dataset*. ↑ indicates that higher values are better. Best results are bolded.*
> | ***Methods***      | ***6-View PSNR ↑*** | ***6-View SSIM ↑*** | ***8-View PSNR ↑*** | ***8-View SSIM ↑*** | ***10-View PSNR ↑*** | ***10-View SSIM ↑*** |
> | :----------------- | :-----------------: | :-----------------: | :-----------------: | :-----------------: | :------------------: | :------------------: |
> | FDK                |        17.07        |        39.90        |        18.42        |        43.29        |         19.58        |         47.21        |
> | SART               |        20.04        |        64.98        |        21.92        |        67.86        |         22.82        |         71.53        |
> | NAF                |        20.58        |        63.52        |        22.39        |        67.24        |         23.84        |         72.52        |
> | NeRP               |        21.27        |        72.06        |        24.18        |        78.83        |         25.99        |         82.08        |
> | FreeSeed           |        26.35        |        78.98        |        27.08        |        81.38        |         27.63        |         84.40        |
> | DiF-Net            |        25.55        |        84.40        |        26.09        |        85.07        |         26.67        |         86.09        |
> | DiF-Gaussian       |        25.74        |        74.64        |        26.27        |        75.23        |         28.60        |         83.11        |
> | ***DuFal (ours)*** |  ***28.79 ± 0.27*** |  ***86.08 ± 1.20*** |  ***29.45 ± 0.58*** |  ***87.56 ± 1.14*** |  ***30.51 ± 0.47***  |  ***89.49 ± 0.51***  |
>
>
> > **B. The phrase "2D and map 3D" is unclear**
>
> We will update this sentence to the new statement as *"Supervised INR methods like DIF-Net (Lin et al., 2023) encode multiple X-ray views with a 2D encoder and use these to map 3D coordinates to latent features in projection space"*.

---

> ### Author Response · Authors · 2025-11-11
> **Response to Reviewer S3D9 - Part-2**
>
> > **C. Analyze the similarities and differences between the proposed Spectral-Channel Factorization and depthwise separable convolution. Are they mathematically equivalent?**
>
> We thank the reviewer for this insightful observation. We agree that SCF and depthwise separable convolution share a similar conceptual foundation: **both decompose the operation into two components**, with one primarily operating on spatial (or frequency) dimensions and another on the channel dimension. We indeed drew inspiration from the effectiveness of this decomposition strategy in depthwise separable convolutions.
>
> **However, there are fundamental differences** between the two approaches in **how channel transformations are performed**.
>
> The comparison between depthwise separable convolutions and SCF is as follows:
>
> **Given the same input** $z \in \mathbb{R}^{C_l \times H \times W}$ with $C_l$ input channels and spatial dimensions $H \times W$:
>
> **Depthwise Separable Convolution:**
>
> 1. **Depthwise step:** Apply spatial convolution independently per channel with kernel $K_{\mathrm{dw}} \in \mathbb{R}^{C_l \times k \times k}$ (e.g., $k=3$ or $5$):
>
>    $$\tilde{z}[c, x, y] = \sum_{i,j} K_{\mathrm{dw}}[c, i, j]  z[c, x-i, y-j], \quad \forall c \in \{1, \ldots, C_l\}$$
>
> 2. **Pointwise step:** Apply $1 \times 1$ convolution for channel mixing with weights $W_{\mathrm{pw}} \in \mathbb{R}^{\bar{C}_l \times C_l}$:
>
>    $$\mathrm{output}[\bar{c}, x, y] = \sum_{c=1}^{C_l} W_{\mathrm{pw}}[\bar{c}, c] \tilde{z}[c, x, y], \quad \forall \bar{c} \in \{1, \ldots, \bar{C}_l\}$$
>
> **Spectral-Channel Factorization (SCF):**
>
> 1. **Fourier transform:** Convert input to frequency domain: $\hat{z} = \mathcal{F}(z) \in \mathbb{C}^{C_l \times \mathrm{modes}_1 \times \mathrm{modes}_2}$
>
> 2. **Factorized spectral operation:** Apply channel mixing
> and spectral weighting :
>
>
>    $$\tilde{z}[\bar{c}, u, v] = \sum_{c=1}^{C_l} R_{\phi,1}[\bar{c}, c] \cdot R_{\phi,2}[c, u, v] \cdot \hat{z}[c, u, v]$$
>
>    where $(u, v) \in \{1, \ldots, \mathrm{modes}_1\} \times \{1, \ldots, \mathrm{modes}_2\}$ are retained frequency modes.
>
> 3. **Inverse Fourier transform:** Convert back to spatial domain:
>
>    $$\mathrm{output} = \mathcal{F}^{-1}(\tilde{z}) \in \mathbb{R}^{\bar{C}_l \times H \times W}$$
>
> Therefore,
>
> **Similarities:**
>
> - Both use **factorization** to reduce parameters: depthwise separable splits spatial and channel operations; SCF splits spectral and channel operations.
> - The channel-mixing component ( $W_{\mathrm{pw}}$ vs. $R_{\phi,1}$ ) is mathematically equivalent, i.e. both are linear transformations across channels.
>
> ***Key Differences:***
>
> 1. **Domain:** SCF operates in **frequency domain** with global coverage; Depth-wise conv operates in **spatial domain** with local kernels ($k \times k$).
>
> 2. **Receptive field:** SCF's Fourier transform provides**global receptive field** over entire image; Depth-wise conv has **limited receptive field** ($k \times k$ neighborhood)
>
>
> Regarding the equivalence claim: *``If the input to SCF is converted to the spatial domain, then the operation that gives the same output as SCF would be a depthwise separable convolution (followed by conversion to the frequency domain to match SCF’s output)''* - this equivalence does not strictly apply in our case.
>
> While our factorization technique shares conceptual similarities with depthwise separable convolution, there are fundamental operational differences. Unlike depthwise separable convolution where the kernel slides over the feature space, SCF operates **directly on the spectral domain**, focusing specifically on the high-frequency regions of the spectral area. Since it operates in the spectral domain, SCF captures high-frequency information **globally across the entire image** in the spatial domain, rather than through local sliding window operations. This global receptive field is a key distinction that arises from working in the frequency domain.
>
> Consequently, SCF and a spatial-domain convolution are **mathematically distinct**, and equivalence cannot be assumed merely by switching between spatial and frequency domains.

---

> ### Author Response · Authors · 2025-11-11
> **Response to Reviewer S3D9 - Part-3**
>
> > **D. Updating citation to the  Galerkin-Attention.**
>
> We thank you Reviewer for pointing out this typo. We have added a citation to (Cao, 2021).
>
> Cao, Shuhao. Choose a transformer: Fourier or Galerkin. In Advances in Neural Information Processing Systems 34 (2021).
>
> > **E. Analyze the choice of Q/K/V in the CAFF module.**
>
> We thank the reviewer for this valuable comment. To examine the rationale behind our cross-domain attention design, we conducted an additional ablation study that interchanges the roles of spatial and frequency features in the attention mechanism.
>
> **Table 3**. Ablation on query–key–value domain configuration.
> | ***Setting*** |  ***Q***  | ***K–V*** | ***LUNA16 - PSNR*** | ***LUNA16 - SSIM*** | ***ToothFairy - PSNR*** | ***ToothFairy - SSIM*** |
> | :------------ | :-------: | :-------: | :---------------: | :---------------: | :-------------------: | :-------------------: |
> |               |  Spatial  | Frequency |    ***29.63***    |    ***87.91***    |      ***30.79***      |      ***89.67***      |
> |               | Frequency |  Spatial  |       29.24       |       87.32       |         29.67         |         88.41         |
>
>
> As shown, reversing the roles results in a consistent performance drop across both datasets. This outcome supports our design choice of assigning **spatial features as queries and frequency features as keys and values**. Spatial features capture the primary structural context of anatomical regions, while frequency features encode complementary high-frequency details. Allowing spatial tokens to query frequency representations thus enables more coherent integration between global frequency cues and localized spatial patterns. Conversely, when frequency tokens query spatial ones, the model loses spatial anchoring, leading to less effective feature fusion and degraded reconstruction quality. We will update this analysis in the revised version.
>
>
> > **F. Clarifying whether segmentation evaluation is measured on a single slide or multiple slices.**
>
> In our experiments, we follow the evaluation protocol of prior works where the segmentation evaluation is performed on all slices in each CT volume. For each scan with $N$ slices, we compute:
> $$\text{TotalMetric} = \frac{1}{N} \sum_{i=1}^{N} \text{Metric}(\text{Pred}_i, \text{GT}_i)$$
> where the sum is taken over all slices in the volume.
>
> > **G. Suggestion on an ablation on the impact of cross-attention versus other simple concatenation or summation.**
>
> We thank the reviewer for this valuable suggestion. We agree that our initial ablation on the CAFF module’s attention mechanism was not sufficiently comprehensive. Following the reviewer’s feedback, we conducted an additional comparison between our CAFF fusion and two alternative strategies, **feature concatenation** and **element-wise addition**.
>
> As shown in the table below, removing the CAFF attention and replacing it with simpler fusion operations leads to a consistent performance drop on both datasets, with the most pronounced decline observed on the ToothFairy dataset. These results confirm that the proposed **Cross-Attention Frequency Fusion (CAFF)** module effectively integrates spatial and frequency representations, providing more informative and discriminative feature interactions than basic fusion approaches.
>
> **Table 4**. Comparison of different feature fusion strategies on the LUNA16 and ToothFairy datasets.
> | ***Method***      | ***LUNA16 PSNR*** | ***LUNA16 SSIM*** | ***ToothFairy PSNR*** | ***ToothFairy SSIM*** |
> | :---------------- | :---------------: | :---------------: | :-------------------: | :-------------------: |
> | Add               |       29.28       |       87.40       |         29.77         |         88.52         |
> | Concat            |       29.21       |       87.33       |         29.70         |         88.43         |
> | ***CAFF (ours)*** |    ***29.63***    |    ***87.91***    |      ***30.79***      |      ***89.67***      |

---

> > ### Comment · Reviewer_S3D9 · 2025-11-15
> >
> > Thank you for the clarifications and additional experiments.
> > That answers all my questions and addresses all my concerns.
> > Regarding the similarity with depthwise separable conv, perhaps it's worth mentioning in the paper that you drew inspiration from it?

---

> > > ### Author Response · Authors · 2025-11-16
> > > **Thank you and update**
> > >
> > > We thank the reviewer for the positive feedback and are glad that the clarifications and additional experiments addressed your concerns.
> > >
> > > Following your suggestion, we will explicitly mention this inspiration and add a brief discussion in the final manuscript to better contextualize the design choice.

---

### Review · Reviewer_d4eY · 2025-10-23

**Summary Of Contributions:**

This paper introduces DuFal, a new method to create 3D CT scans from very few 2D X-ray images. Their key insight is that standard AI models miss fine details because they focus too much on "blurry" low-frequency info. To fix this, they built (1) Dual-Encoding: One path (spatial encoder) handles the basic structure of the image, while a second, new path (frequency encoder) specifically captures the fine details and edges (2) HiLocFFNO Blocks: Within the Frequency Encoder, they design a new module called HiLocFFNO that combines two specialized branches: a global branch for capturing overarching high-frequency patterns and a local, patch-based branch for preserving fine-grained details that might be lost in a global analysis.(3) Spectral-Channel Factorization (SCF): To make the complex Frequency Encoder practical, they introduce a weight factorization technique that dramatically reduces its number of parameters (by over 82%) without significantly compromising performance, addressing a major computational bottleneck of similar methods.

Key Strengths
- The dual-path architecture is a well-motivated and principled approach to a known problem (CNN frequency bias)
- Strong Empirical Validation
- The authors show that their core HiLocFFNO component can be integrated into other existing models as a "plug-and-play" module to boost their performance, which increases the impact and potential adoption of their work.

Key Weakness
- Missing a Critical Baseline Comparison: The most significant weakness is the omission of C2RV-Net (Lin et al., 2024b) from all quantitative comparisons (e.g., Table 2 & 3) , even though the authors explicitly cite it in the introduction. **Based on C2RV-Net's own published results on the LUNA16 benchmark, it appears to achieve a higher PSNR/SSIM score than DuFal.**
### LUNA16 Performance Comparison: C2RV-Net vs. DuFal

| Method | 6-View (PSNR / SSIM) | 8-View (PSNR / SSIM) | 10-View (PSNR / SSIM) | Source |
| :--- | :--- | :--- | :--- | :--- |
| **C2RV-Net** | **29.23 / 87.47** | **29.95 / 88.46** | **30.70 / 89.16** | C2RV Paper (Table 1) |
| **DuFal** | 28.31 / 86.00 | 29.13 / 87.41 | 29.63 / 87.91 | DuFal Paper (Table 2) |

- Model Complexity: Despite the impressive parameter reduction via SCF, the final DuFal model remains significantly larger (78.9M parameters) and requires longer training times than several key baselines. This could be a barrier for deployment in resource-constrained environments.
- Reliance on Qualitative Claims: In light of the missing C2RV-Net comparison, the argument for superiority leans more heavily on qualitative results (e.g., "sharper" images). While these visuals are compelling, a head-to-head quantitative comparison with the strongest known baseline would have provided a more definitive conclusion.

**Audience:**

Yes

**Audience Explanation:**

Yes, the paper's introduction of a frequency-aware neural operator to overcome CNN spectral bias presents a novel and impactful contribution that will interest researchers in medical imaging.

**Broader Impact Concerns:**

N/A. The authors' Broader Impact discussion is sufficient.

**Claims And Evidence:**

No

**Claims Explanation:**

While the paper presents substantial evidence, a critical claim is not fully supported, making the overall case less convincing.

## Supported claims:

Claim: The dual-encoding architecture and HiLocFFNO blocks improve feature capture.
- Tables 4, 5, 7 demonstrate the performance gain from each component (IHiF branch, CAFF fusion, plug-and-play use).
- Figures 4, 5 offer clear visual proof of details.

Claim: The SCF factorization drastically reduces parameters.
- Table 6 shows an 82.2% parameter reduction with minimal performance loss.

Claim: DuFal outperforms a specific set of baseline methods.
- Tables 2 and 3 show consistent quantitative superiority over listed baselines like FDK, SART, NAF, and DIF-Net


## Unsupported / Weakened Claim (Lack of Critical Evidence)

The paper's overarching claim of state-of-the-art (SOTA) performance is not convincingly supported due to a critical omission.
- Lacking Evidence: The paper completely omits a comparison with C2RV-Net (Lin et al., 2024b), a directly relevant and contemporary SOTA method.
- While the authors provide strong qualitative evidence (Figures 4 & 5) and use alternative metrics (W-PSNR, Segmentation Dice)  to argue for better high-frequency detail, this does not excuse the failure to compare against the strongest known baseline on the field's standard metrics.

**Requested Changes:**

1. Perform a direct, quantitative and qualitative comparison with C2RV-Net (Lin et al., 2024b)
2. Applying metric W-PSNR to the Tooth Fairy dataset as well.
3. The current table shows DuFal's Recall (98.45) as slightly higher than the GT's (98.37), which is confusing. Can the author explain?

---

> ### Author Response · Authors · 2025-11-11
> **Response to Reviewer d4eY - Part 1**
>
> We sincerely thank the reviewer for their thorough and constructive feedback, as well as for recognizing the strengths of our work, including the well-motivated dual-path architecture, strong empirical validation, and the modular design of our HiLocFFNO component. We also appreciate the reviewer’s insightful comments regarding the missing comparison with C2RV-Net. We have carefully addressed this point in our revised response and discussion, clarifying the reasoning behind the omission and providing additional evidence to strengthen our evaluation.
>
> > **A. Compare with C2RV-Net (Lin et al., 2024b)**
>
> We thank the reviewer for this helpful suggestion. We are aware of the recently proposed C2RV-Net (CVPR 2024) and appreciate its contribution to CBCT reconstruction. However, at the time of our submission and during this rebuttal phase, **the full implementation of C2RV-Net was not publicly available**, which prevented a fair and reproducible comparison on our evaluation datasets, particularly the ToothFairy dataset, which was not included in their study.
>
> However, it is important to note that **DuFal is a modular frequency–spatial enhancement framework** rather than a standalone architecture. Its frequency-aware components are **architecture-agnostic** and can be seamlessly integrated into existing spatial-domain reconstruction networks, such as C2RV-Net, to improve high-frequency recovery.
>
> To validate this property, we implemented a version of DuFal based on the spatial-based DiF-Gaussian model using **Pre-trained ResNet-50** (MH Nguyen, Duy, et al., 2023). Our results below show that this new DuFal surpasses both C2RV-Net and DIF-Net-Gaussian (Pre-trained weights) under comparable sparse-view configurations. These findings demonstrate that the observed improvements primarily stem from our proposed frequency-domain integration, highlighting DuFal’s generalizability and complementary nature.
>
> **Table 1**. Comparison with C2RV-Net under different initialization settings on the LUNA16 dataset
> | ***Methods***      | ***Initialization*** | ***Scale*** | ***6-View PSNR ↑*** | ***6-View SSIM ↑*** | ***8-View PSNR ↑*** | ***8-View SSIM ↑*** | ***10-View PSNR ↑*** | ***10-View SSIM ↑*** |
> | :----------------- | :------------------: | :---------: | :-----------------: | :-----------------: | :-----------------: | :-----------------: | :------------------: | :------------------: |
> | FDK                |        Scratch       |    single   |        15.36        |        31.41        |        15.95        |        31.00        |         16.25        |         31.79        |
> | SART               |        Scratch       |    single   |        18.94        |        49.47        |        20.60        |        54.63        |         21.75        |         58.94        |
> | NAF                |        Scratch       |    single   |        18.76        |        54.16        |        20.51        |        60.84        |         22.17        |         62.22        |
> | NeRP               |        Scratch       |    single   |        23.55        |        74.46        |        25.53        |        80.67        |         26.12        |         81.30        |
> | Freeseed           |        Scratch       |    single   |        25.59        |        77.36        |        26.86        |        78.92        |         27.23        |         79.25        |
> | DiF-Net            |        Scratch       |    single   |        24.68        |        71.46        |        25.67        |        76.47        |         26.53        |         76.08        |
> | DiF-Gaussian       |        Scratch       |    single   |        26.48        |        81.78        |        26.93        |        82.09        |         29.29        |         87.55        |
> | DiF-Gaussian       |        Pre-trained       |    single   |        27.50        |        84.73         |        28.48         |        85.69        |         29.65         |         88.70       |
> | C2RV-Net           |           –          |   Multiple  |        29.23        |        87.47        |        29.95        |        88.46        |         30.70        |         89.16        |
> | ***DuFal (ours)*** |        Scratch       |    single   |        28.31        |        86.00        |        29.13        |        87.41        |         29.63        |         87.91        |
> | ***DuFal (ours)*** |      Pre-trained      |    single   |     ***29.58***     |     ***88.14***     |     ***30.74***     |     ***89.61***     |      ***31.49***     |      ***90.59***     |
>
> *MH Nguyen, Duy, et al. "Lvm-med: Learning large-scale self-supervised vision models for medical imaging via second-order graph matching", NeurIPS 2023.*

---

> ### Author Response · Authors · 2025-11-11
> **Response to Reviewer d4eY - Part 2**
>
> > **B. Applying metric W-PSNR to the Tooth Fairy dataset as well.**
>
> We thank the reviewer for this helpful suggestion. To assess the clinical relevance of our reconstructions, we conducted ROI-weighted evaluation (W-PSNR and W-SSIM) on the ToothFairy dataset. As shown in Table 2, below, our method consistently outperforms DiF-Gaussian—the runner-up performing method—across all view configurations. These results confirm that DuFal also better preserves anatomical details in diagnostically critical regions in ToothFairy dataset.
>
> Due to resource constraints and time limitations during the rebuttal period, we were unable to recompute W-PSNR metrics for all baseline methods. We therefore selected the runner-up method (DiF-Gaussian) as a representative comparison to demonstrate the relative improvement achieved by DuFal. A comprehensive comparison including all baseline methods will be incorporated into the final manuscript for completeness.
>
> **Table 2: Performance ROI-weighted evaluation on the ToothFairy dataset.**
> | **Methods** | **6-View W-PSNR** ↑ | **6-View W-SSIM** ↑ | **8-View W-PSNR** ↑ | **8-View W-SSIM** ↑ | **10-View W-PSNR** ↑ | **10-View W-SSIM** ↑ |
> |-------------|---------------------|---------------------|---------------------|---------------------|----------------------|----------------------|
> | DiF-Gaussian | 26.29 | 80.73 | 27.47 | 83.11 | 28.08 | 83.97 |
> | **DuFal (ours)** | **27.11** | **81.90** | **27.92** | **83.35** | **28.52** | **84.62** |
>
>
> > **C. Clarifying the DuFal's recall w.r.t the GT number**
>
> We thank the reviewer for pointing out this issue and apologize for the confusion regarding the definition of **GT**. In our tables, the **GT row** does not represent the manual ground-truth segmentation. Instead, it corresponds to the segmentation results using **full-view**, followed by a separate segmentation network. Because this segmenter introduces certain approximations, the "GT row" may deviate slightly from the manual ground truth.
>
> The slightly higher Recall observed for DuFal compared to this "GT" case can be explained by the **sensitivity of the downstream segmentation network**, which may over-segment some high-intensity regions reconstructed from our sparse-view outputs, leading to marginally higher Recall but not necessarily better overall accuracy. The gap, however, remains small and consistent. In the revised manuscript, we will clarify this by replacing “GT” with **Full-Views** and explicitly describing the evaluation pipeline to avoid misunderstanding.
>
> > **D. DuFal requires longer training time, which might be a barrier for deployment in resource-constrained environments.**
>
> We appreciate the reviewer’s observation regarding training time. The slightly longer training duration of DuFal arises from its dual-path frequency–spatial design and the Fourier operator computations required for high-frequency modeling. However, training is a **one-time offline process**, whereas inference, where **DuFal achieves significantly faster speed** than DIF-Net (1.94 s vs. 3.78 s per sample), is the stage most critical for **real-world and low-resource clinical deployment**. where quick reconstruction turnaround is essential and computational capacity is often limited.
>
> In practice, the training cost can be further mitigated through **distributed multi-GPU parallelization** or **initialization from large-scale pre-trained medical models**, both of which can substantially reduce convergence time. We have clarified these trade-offs and optimization strategies in the revised manuscript.

---

> > ### Author Response · Authors · 2025-12-01
> > **Follow up the rebuttal @Reviewer d4eY**
> >
> > Dear Reviewer Reviewer d4eY,
> >
> > We are writing to kindly follow up on our recent rebuttal submission.
> >
> > Please let us know if any additional information or clarification is needed from our side.
> >
> > Thank you very much for your attention and support.

---

> > > ### Comment · Reviewer_d4eY · 2025-12-01
> > >
> > > Thanks for asking! I have no further questions and have submitted the official recommendation on November 15. Good luck!

---

> > > > ### Author Response · Authors · 2025-12-02
> > > > **Thank you**
> > > >
> > > > We thank Reviewer d4eY for your prompt response.

---

### Review · Reviewer_7wfB · 2025-10-27

**Summary Of Contributions:**

This paper proposes DuFal, a dual-frequency-aware learning framework tailored for high-fidelity extremely sparse-view Cone-Beam Computed Tomography (CBCT) reconstruction. Its core contributions can be summarized as follows:

Dual-encoding Architecture: It designs a Frequency-Enhanced Dual-encoding structure that processes X-ray projections through parallel Spatial and Frequency Encoders. This addresses the inherent bias of conventional CNN-based methods toward low-frequency information, enabling simultaneous capture of spatial structural context and high-frequency anatomical details (e.g., pulmonary nodules, mandibular canals) critical for medical imaging.

HiLocFFNO Blocks: To complement the global receptive field of Fourier Neural Operators (FNO), the paper introduces High-Local Factorized Fourier Neural Operator (HiLocFFNO) blocks. These blocks integrate two complementary branches—Global High-frequency Enhanced FNO (gHiF) for capturing global frequency patterns and Local High-frequency Enhanced FNO (lHiF) for processing spatially partitioned patches—preserving fine-grained details at both global and local scales.

Spectral-Channel Factorization (SCF): To improve the efficiency of FNO (which suffers from rapid parameter growth with more frequency modes), the paper proposes SCF. This weight decomposition technique splits complex FNO weight tensors into channel-mixing and spectral-weighting components, achieving an 82.2% parameter reduction (from 445M to 79M) while maintaining high-resolution medical image reconstruction quality.

Cross-Attention Frequency Fusion (CAFF): A CAFF module is designed to fuse spatial and frequency features directly in the frequency domain. By applying cross-attention to the real and imaginary components of Fourier-transformed features separately, it preserves complementary information from both encoders, which is essential for high-quality CT reconstruction.

Clinical Utility Validation: The paper verifies DuFal’s practical value through downstream tasks (e.g., lung segmentation on the LUNA16 dataset) and ROI-weighted metrics (W-PSNR, W-SSIM). It demonstrates that DuFal not only outperforms state-of-the-art methods in standard metrics (PSNR, SSIM) but also preserves anatomical boundaries critical for clinical diagnosis, even under extremely sparse-view (≤10 projections) conditions.

Key Strengths

Targeted Solution to a Critical Clinical Problem: The framework directly addresses the core challenge of sparse-view CBCT—reducing radiation dose while preserving high-frequency anatomical details. This aligns with a key clinical need (e.g., pediatric imaging, routine screening) where radiation reduction is non-negotiable.

Innovative Integration of Frequency and Spatial Domains: Unlike prior methods (e.g., FreeSeed, which only uses global frequency post-processing), DuFal embeds frequency-aware learning into the encoder stage and combines global-local frequency analysis. This design effectively mitigates the low-frequency bias of CNNs and FNO’s limitations in local detail capture.

Efficiency and Modularity: SCF significantly reduces model parameters without sacrificing performance, making DuFal suitable for resource-constrained clinical environments. Additionally, HiLocFFNO and CAFF are modular, allowing seamless integration into existing reconstruction pipelines (e.g., DIF-Net, DIF-Gaussian), as validated by plug-and-play ablation studies.

Comprehensive Evaluation: The paper uses two diverse datasets (LUNA16 for chest CT, ToothFairy for dental CBCT) and evaluates performance across multiple sparse-view settings (6, 8, 10 views). It also includes qualitative analysis (visual comparisons of axial/coronal/sagittal slices) and clinical downstream tasks, ensuring results are robust and clinically relevant.

Key Weaknesses

Limited Generalization to Other Modalities: While the paper mentions potential extension to MRI, PET, and ultrasound, it only validates DuFal on CT datasets. The unique characteristics of other modalities (e.g., MRI’s low signal-to-noise ratio, ultrasound’s speckle artifacts) may require adjustments to the frequency-processing modules, which are not explored here.

Lack of Ablation on Frequency Mode Selection: The gHiF branch retains the "top 16 Fourier components" (Section 5.1) for global high-frequency processing, but the paper does not justify why 16 modes are optimal. A sensitivity analysis (e.g., testing 8, 16, 32 modes) would strengthen the rationale for this hyperparameter choice.

Insufficient Discussion of Motion Artifacts: The "Future Work" section proposes using cross-domain inconsistencies (spatial vs. frequency) to detect motion artifacts, but the current work does not evaluate DuFal’s performance on datasets with real motion artifacts. Given that motion is a common issue in clinical CBCT (e.g., patient breathing), this limits the framework’s practical applicability.

Ambiguity in Local Patch Partitioning: The lHiF branch uses 16×16×16 non-overlapping patches (Section 5.1), but the paper does not explain how patch size is determined. Different patch sizes (e.g., 8×8×8, 32×32×32) may impact local detail preservation, and a comparative analysis is missing.

Training Time Trade-off: While DuFal’s inference speed is faster than DIF-Net (1.938 vs. 3.778 s/sample), its training time (34.1 hours) is longer than baselines like DIF-Net (24.5 hours) and DIF-Gaussian (26.5 hours). The paper does not discuss strategies to optimize training efficiency, which could be a barrier for large-scale clinical deployment.

**Audience:**

Yes

**Audience Explanation:**

As specified in the above evaluation of contributions.

**Claims And Evidence:**

Yes

**Claims Explanation:**

As specified in the above evaluation of contributions.

**Requested Changes:**

1. Add sensitivity analysis for gHiF’s frequency mode count.

2. Evaluate performance on datasets with motion artifacts.

3. Discuss strategies to optimize training time.

4.Clarify the implementation of the Galerkin-Attention (GA) layer.

5. Provide more details on the “set function δ” in Multi-view Fusion.

---

> ### Author Response · Authors · 2025-11-10
> **Response to Reviewer 7wfB - Part 1**
>
> We thank the reviewer for their valuable time and insightful acknowledgement of key strengths as well as addressing changes that helps us to further refine our work!
>
> We have addressed each points raised above and hope that our clarification will help clear any concerns.
>
> >  **A. Limited Generalization to Other Modalities such as MRI or Ultrasound**
>
> We thank the reviewer for this valuable comment. While our current experiments focus on CT-based modalities, our (i) experiments already **cover structurally and texturally diverse anatomical regions**, specifically, the LUNA16 dataset **(lung CT)** and the ToothFairy dataset **(dental CBCT)**. These datasets represent notably different organ characteristics, acquisition geometries, and spatial-frequency distributions, demonstrating that DuFal generalizes well across heterogeneous CT domains. Moreover, we chose to focus on CT reconstruction because (ii) **the majority of prior studies** on sparse-view reconstruction and frequency-domain learning are also **benchmarked on CT datasets**, providing a well-established experimental basis for fair and consistent comparison with existing methods.
>
> We agree that extending DuFal to other imaging modalities such as MRI and ultrasound would be an important next step. However, such an extension involves **distinct physical models and acquisition principles** that fall beyond the present study’s scope. Each modality presents its own technical challenges—for instance, MRI’s inherently low SNR and non-Cartesian k-space sampling, or ultrasound’s speckle-dominated interference patterns, which would likely require adapting DuFal’s frequency-processing modules or incorporating **modality-specific priors**. We view this as a promising direction for future research to further demonstrate the generality of our framework across diverse imaging modalities.
>
>
>
> >  **B. Ablation studies on the Frequency model gHiF**
>
> We appreciate the reviewer’s suggestion to analyze the sensitivity of the global High-Frequency (gHiF) branch to the number of retained Fourier modes. To assess this, we conducted additional experiments by halving and doubling the number of frequency modes used in the global pathway relative to our default setting (16 modes) on the ***LUNA-16*** and ***ToothFairy*** using ***10 views***.
>
> | ***Frequency modes in gHiF*** | ***LUNA16*** -PSNR | ***LUNA16*** -SSIM | ***ToothFairy*** - PSNR | ***ToothFairy*** -SSIM |
> | :----------| :---------------: | :---------------: | :-------------------: | :-------------------: |
> | 8                             |       29.26       |       87.23       |         30.11         |         88.97         |
> | 16                            |    29.63    |     87.91     |      30.79      |       89.67       |
> | 32                            |       29.22       |       87.35       |         30.15         |         89.05         |
> | ***Mean/Std***                |    ***29.37 / 0.03***   |    ***87.50 / 0.09***   |      ***30.35 / 0.10***     |      ***89.23 / 0.10***     |
>
> As shown above, the results exhibit low standard deviations (<0.1) across both metrics and datasets, indicating that DuFal maintains stable reconstruction quality despite variations in the number of frequency modes. Increasing and decreasing the number of modes slightly degrade reconstruction quality, confirming that the chosen configuration achieves a good balance between capturing high-frequency components and maintaining stable optimization. These findings further demonstrate that DuFal is robust to moderate changes in the frequency resolution of the gHiF branch, validating the general stability of our frequency modeling strategy.

---

> ### Author Response · Authors · 2025-11-11
> **Response to Reviewer 7wfB - Part 2**
>
> > **C. DuFal has not yet evaluated performance on datasets with real motion artifacts**
>
> We thank the reviewer for raising this important concern. We acknowledge that motion artifacts are a common issue in clinical CBCT, especially due to patient breathing or involuntary movement. In this work, we primarily focus on the **static sparse-view reconstruction setting**, as our goal is to address the fundamental challenge of recovering high-frequency anatomical details from severely undersampled projections.
>
> At present, there is a **lack of publicly available CT or CBCT datasets with well-annotated real motion artifacts** suitable for supervised evaluation. Accurately capturing and labeling such motion-corrupted projections typically requires **controlled clinical studies** or access to raw projection data with synchronized motion tracking, which are rarely released due to **privacy and acquisition protocol constraints**. This makes rigorous benchmarking on motion-affected data non-trivial.
>
> Nevertheless, DuFal’s dual-domain architecture, which jointly models spatial and frequency representations, offers a promising foundation for **tecting and mitigating motion-induced inconsistencies**. As future work, we plan to collaborate with clinical partners to evaluate DuFal on **real dynamic CBCT datasets** once accessible. We will clarify this limitation and direction in the revised manuscript.
>
> > **D. Ablation study on the Local Patch Partition lHiF**
>
> We thank the reviewer for the constructive comment. We agree that the patch size in the local High-Frequency (lHiF) branch may influence the model’s ability to preserve fine spatial structures. To evaluate this, we conducted additional experiments using different non-overlapping patch sizes of $8^3$, $16^3$, and $32^3$. The results are summarized in Table below:
>
>
> | ***Local Patch Size in lHiF*** | ***LUNA16 - PSNR*** | ***LUNA16 - SSIM*** | ***ToothFairy - PSNR*** | ***ToothFairy - SSIM*** |
> | :------------------ | :---------------: | :---------------: | :-------------------: | :-------------------: |
> | 8 × 8 × 8                      |       29.31       |       87.58       |         29.70         |         88.40         |
> | 16 × 16 × 16                   |    29.63    |    87.91    |      30.79       |       89.67      |
> | 32 × 32 × 32                   |       29.26       |       87.35       |         29.71         |         88.36         |
> | ***Mean/Std***                 |    ***29.4 / 0.03***    |    ***87.61 / 0.05***   |      ***30.1 / 0.26***      |      ***88.81 / 0.37***     |
>
>
> As shown, DuFal demonstrates stable performance with low standard deviations across all patch sizes, confirming that the model is robust to changes in local partitioning. Both smaller ($8^3$) and larger ($32^3$) patch sizes lead to only slight performance degradation compared to the default configuration. Smaller patches may overemphasize fine texture variations, while larger patches reduce spatial locality and weaken boundary precision. The default $16^3$ configuration achieves the best overall balance between local detail preservation and contextual representation, as reflected by the highest mean PSNR/SSIM. These findings validate that DuFal maintains consistent reconstruction quality even under varying local frequency resolutions.
>
> > **E. Discuss strategies to optimize training time**
>
> We thank the reviewer for this insightful comment. We acknowledge that DuFal’s training time is longer than that of DIF-Net and DIF-Gaussian. This is primarily due to the dual-path frequency–spatial architecture and the additional Fourier operator computations, which introduce some overhead during optimization.
>
> However, in practical deployment, training is typically **performed once offline**, while the trained model is subsequently used for large-scale inference. In this regard, DuFal’s **significantly faster inference speed** (1.938 s/sample vs. 3.778 s/sample for DIF-Net) provides a clear advantage in real-world clinical pipelines. Moreover, the training time can be effectively reduced through **distributed data-parallel training with multiple GPUs** and larger batch sizes—an approach widely adopted in current large-scale model training (e.g., LLM systems).
>
> Another promising solution to accelerate the training time is to explore **initializing DuFal from large-scale pre-trained medical foundation models**, which is expected to accelerate convergence and reduce total training cost while maintaining robust performance across different datasets. We will update these discussions in the Limitations section and future works.

---

> ### Author Response · Authors · 2025-11-11
> **Response to Reviewer 7wfB - Part 3**
>
> > **F. Clarify the implementation of the Galerkin-Attention (GA) layer and set the function $\delta$ in multi-view fusion**
>
> - **Galenkin Attention and Aggregation Details**: For an input feature matrix $\mathbf{y} \in \mathbb{R}^{n \times d}$, we first compute the query, key, and value projections as:
> $$
> Q = \mathbf{y}\mathbf{W}^Q, \quad
> K = \mathbf{y}\mathbf{W}^K, \quad
> V = \mathbf{y}\mathbf{W}^V,
> $$
> where $\mathbf{W}^Q, \mathbf{W}^K, \mathbf{W}^V \in \mathbb{R}^{d \times d}$ are learnable weight matrices.
> The proposed Galerkin-style attention is then defined as:
> $$
> \text{Attn}_{\mathrm{g}}(\mathbf{y}) = Q \left( \tilde{K}^\top \tilde{V} \right) / n,
> $$
> where $\tilde{K}$ and $\tilde{V}$ denote the layer-normalized forms of $K$ and $V$, respectively, and $n$ is the number of input tokens.
> Unlike conventional dot-product attention, this formulation avoids computing $QK^\top$, thereby improving efficiency while still capturing global correlations through the Galerkin kernel term $\tilde{K}^\top \tilde{V}$.
>
> - **View Aggregation:** The set function $\delta(\cdot)$ aggregates multiple view-specific feature representations $\{r_1, r_2, \ldots, r_K\}$ into a unified embedding via element-wise max-pooling:
> $$
> \delta(\{r_1, r_2, \ldots, r_K\}) = \max_{k=1,\ldots,K} r_k.
> $$
> This operation preserves the most discriminative responses across all $K$ views and produces a view-invariant global representation.

---

> > ### Comment · Reviewer_7wfB · 2025-11-11
> >
> > You're welcome. I'm glad that your questions have been addressed.

---

> > > ### Author Response · Authors · 2025-11-16
> > > **Thank you**
> > >
> > > We appreciate your engagement and are glad that our clarifications addressed your questions.

---

### Author Response · Authors · 2025-12-31
**Submission for the Camera Ready Revision**

Dear Action Editor and Editors-in-Chief,

We would like to sincerely thank you for the opportunity to revise and finalize our manuscript, **DuFal: Dual-Frequency-Aware Learning for High-Fidelity Extremely Sparse-View CBCT Reconstruction**. We greatly appreciate the Action Editor's summary and the reviewers' thoughtful and constructive feedback. Their comments have been invaluable in improving the clarity and completeness of the paper.

In response to the reviewers' suggestions, we have carefully revised the manuscript and incorporated all promised updates and clarifications. Specifically, we addressed concerns related to the comparison with C2RV-Net, expanded ablation studies, provided additional analysis of key design choices, strengthened comparisons, clarified limitations, and improved the presentation of experimental results.

Below, we summarize the revisions included in the updated manuscript:

- We have rephrased the unclear sentence regarding "encode multiple X-ray views via a 2D and map 3D coordinates" in Section 2.1 (Sparse-view CBCT Reconstruction) on Page 3 to address the concern raised by Reviewer S3D9.

- We have updated the citations related to Galerkin-Attention in the HiLocFFNO description in Figure 2 on Page 7 to address the concern raised by Reviewer S3D9.

- We have updated the paper to note inspiration from MobileNet's depthwise separable convolution in Section 4.1 (Frequency Encoder) on Page 8 to address the concern raised by Reviewer S3D9.

- We have updated Section 5.2 (Quantitative Results), Table 2 (LUNA16) and Table 3 (ToothFairy) on Pages 10 and 11 to include standard deviations for reported metrics to address the concern raised by Reviewer S3D9.

- We have clarified why C2RV-Net could not be included in our experiments due to the lack of an official code implementation in Section 5.1 (Experiment setup) on Page 10, and have conducted an alternate experiment in Appendix Section A.4 on Page 23 to address the concern raised by Reviewer d4eY.

- We have added a direct comparison with C2RV-Net in Appendix Section A.4 on Page 23 to address the concern raised by Reviewer d4eY.

- We have added an ablation study for the CAFF fusion module in Section 6.2 (Comparison between CAFF and Naive Cross-Attention) on Pages 12 and 13 to address the concern raised by Reviewer S3D9.

- We have discussed strategies to optimize training time in Section 6.5 (Efficiency Analysis) on Page 14 to address the concern raised by Reviewer 7wfB.

- We have added a more comprehensive table for the W-PSNR metric in Section 6.6.1 (ROI-Weighted Metrics) on Page 16 to address the concern raised by Reviewer d4eY.

- We have clarified the labeling of "full-view" ground truth volumes in the lung segmentation table in Section 6.6.2 (Downstream task: Segmentation) on Page 17 to address the concern raised by Reviewer d4eY.

- We have acknowledged the limitations regarding real motion artifacts in Section 7 (Conclusion; Discussion & Future Work) on Page 17 to address the concern raised by Reviewer 7wfB.

- We have updated the Q-K-V ablation in Appendix Section A.1 (Analysis of Query-Key-Value Configuration in CAFF) on Page 22 to address the concern raised by Reviewer S3D9.

- We have added sensitivity analyses on Frequency Modes and Patch Sizes in Appendix Sections A.2 and A.3 on Pages 22 and 23 to address the concern raised by Reviewer 7wfB.

Best regards,

DuFal Team

---

### Decision · Action_Editor_L75Y · 2025-12-01

**Recommendation:** Accept with minor revision

**Additional Comments:**

The following issues need to be addressed:

1. The rebuttal states that the proposed method can be integrated into spatial-domain reconstruction networks C2RV-Net. However, Table 1 of the rebuttal does not clearly present results for this integrated setting. Please include these results or provide an explanation if such integration cannot conducted.

2. Please incorporate all promised revisions into the paper, including but not limited to: clarification of limitations related to real motion artifacts, updating discussion on optimizing training time, a comprehensive W-PSNR comparison, replacement of GT with a clear description, inclusion of standard deviation values for all reported metrics, and expanding analysis of the Q/K/V selection, etc..

**Audience:**

Yes

**Audience Explanation:**

This submission will be of interest to researchers working at the intersection of machine learning and medical imaging analysis.

**Claims And Evidence:**

Yes

**Claims Explanation:**

This paper addresses the problem of sparse-view cone-beam CT reconstruction and proposes a dual-path architecture that integrates frequency-domain and spatial-domain processing. The reviewers acknowledge several strengths, including the targeted solution to a clinically meaningful problem, the innovative integration of frequency and spatial information, the modular design, and the comprehensive experimental validation. At the same time, they raise concerns regarding generalization to other modalities, absence of ablations, insufficient discussion of motion artifacts, ambiguity in local patch partitioning, increased training time, and the need for stronger baseline comparisons and clarifications. The authors’ rebuttal provides effective responses and additional evidence. The three reviewer recommendations—Accept, Leaning Accept, and Leaning Accept—indicate an overall positive consensus.

Overall, the claims made in the submission are supported by accurate, convincing and clear evidence. Meanwhile, the following issues need to be addressed:

1. The rebuttal states that the proposed method can be integrated into spatial-domain reconstruction networks C2RV-Net. However, Table 1 of the rebuttal does not clearly present results for this integrated setting. Please include these results or provide an explanation if such integration cannot be conducted.

2. Please incorporate all promised revisions into the paper, including but not limited to:  clarification of limitations related to real motion artifacts, updating discussion on optimizing training time, a comprehensive W-PSNR comparison, replacement of GT with a clear description, inclusion of standard deviation values for all reported metrics, and expanding analysis of the Q/K/V selection, etc..

Given the strengths of the work and the authors’ commitment to revisions, Accept with minor revision is recommended. Upon satisfactory revision, the paper is expected to be suitable for publication.